# Predominance of *cis*-regulatory changes in parallel expression divergence of sticklebacks

**Jukka-Pekka Verta[1,2]\*, Felicity C Jones[1]\***

[1]Friedrich Miescher Laboratory of the Max Planck Society, Max-Planck-Ring, Tübingen, Germany; [2]Organismal and Evolutionary Biology Research Programme, University of Helsinki, Helsinki, Finland

**Abstract** Regulation of gene expression is thought to play a major role in adaptation, but the relative importance of *cis*- and *trans*- regulatory mechanisms in the early stages of adaptive divergence is unclear. Using RNAseq of threespine stickleback fish gill tissue from four independent marine-freshwater ecotype pairs and their F1 hybrids, we show that *cis*-acting (allele-specific) regulation consistently predominates gene expression divergence. Genes showing parallel marine-freshwater expression divergence are found near to adaptive genomic regions, show signatures of natural selection around their transcription start sites and are enriched for *cis*-regulatory control. For genes with parallel increased expression among freshwater fish, the quantitative degree of *cis*- and *trans*-regulation is also highly correlated across populations, suggesting a shared genetic basis. Compared to other forms of regulation, *cis*-regulation tends to show greater additivity and stability across different genetic and environmental contexts, making it a fertile substrate for the early stages of adaptive evolution.

DOI: https://doi.org/10.7554/eLife.43785.001

**\*For correspondence:**
jukka-pekka.verta@helsinki.fi (J-PV);
fcjones@tuebingen.mpg.de (FCJ)

**Competing interests:** The authors declare that no competing interests exist.

## Introduction

The ability of organisms to rapidly adapt to new environments can be both facilitated and constrained by the underlying molecular basis and mechanisms operating at the genomic level. There have been significant advances in our understanding of the genomic basis of adaptive evolution including that adaptation is often polygenic and involves loci that are predominantly intergenic and putatively regulatory (*Brawand et al., 2014*; *Grossman et al., 2013*; *Jones et al., 2012*). Transcriptional regulation of gene expression can be controlled through *cis*-acting regulatory elements that are linked to their target gene alleles (e.g. promotors, enhancers), or through *trans*-acting mechanisms such as transcription factors whose action typically impacts both target alleles.

During rapid adaptation, selection may favor master regulator genes that trigger concerted changes in many downstream genes within a gene regulatory network through *trans*-acting mechanisms. This view has theoretical backing and has been demonstrated in multiple examples (*Cooper et al., 2003*; *Stern and Orgogozo, 2009*). Alternatively, selection may favor the modularity and tighter linkage offered by *cis*-acting mechanisms under the following scenarios: moderate changes to single alleles in a tissue-specific manner, rather than systemic changes to gene expression; or frequent out-crossing or hybridisation (*Carroll, 2008*). *Cis*- and *trans*-regulatory mechanisms are not mutually exclusive and adaptation is expected to promote co-evolution between *cis*- and *trans*-acting mechanisms so that optimal gene expression levels are reached and maintained (*Fraser et al., 2010*). Interdependence of *cis*- and *trans*-regulatory mechanisms has been hypothesized to act as a barrier for gene flow and contribute to incipient speciation: incompatible regulatory

factors fail to promote optimal gene expression levels in hybrid progeny (*Landry et al., 2007a*; *Landry et al., 2005*).

Given their different properties, it stands to reason that different evolutionary scenarios and selection contexts may alternatively favor *trans-* and *cis-*acting mechanisms in the early stages of intraspecific adaptive divergence (*Coolon et al., 2014*; *Fraser et al., 2010*; *Hart et al., 2018*; *Lemos et al., 2008*; *Stern and Orgogozo, 2009*). In this context, divergent adaptation to local environments often occurs in the face of ongoing gene flow. The shuffling effects of recombination will tend to dissociate co-evolved factors and selection may be more efficient on the larger effect haplotypes carrying multiple such factors that are maintained by linkage-disequilibrium. Thus, the advantage of a rapid adaptive response mediated via a small number of *trans-*regulatory mutations in a gene regulatory network, may shift to favor *cis-*regulatory architecture where co-evolved mutations are more closely linked to each other and the gene whose expression they regulate. Parallel evolution provides a powerful context to explore the relative importance of *cis-* and *trans-* regulation in the early stages of intraspecific adaptive divergence. Using independent biological replicates of the evolutionary process it is possible to ask whether the same phenotype has evolved via the same or different molecular underpinnings. While regulatory changes seem to predominate in adaptation of natural populations we know little about the extent and parallelism in gene expression and its *cis-* and *trans-*regulation.

The threespine stickleback fish is an excellent system to address these questions. Following the retreat of the Pleistocene ice sheet 10–20 k years ago, the parallel evolution of freshwater ecotypes from ancestral marine forms has occurred repeatedly and independently in thousands of populations across the Northern Hemisphere (*Bell and Foster, 1994*). Considerable evidence points toward an important role for gene regulation in this adaptive divergence. Firstly, forward mapping and functional dissection have identified mutations in cis-regulatory elements underlying the parallel loss of major morphological traits (bony armor plates *Colosimo et al., 2005*; *O'Brown et al., 2015*) and pelvic spines *Shapiro et al., 2004*; *Chan et al., 2010*). Further, whole genome sequencing of marine and freshwater sticklebacks from multiple populations revealed that repeated parallel evolution of freshwater ecotypes from marine ancestors involves reuse of pre-existing genetic variation at ~81 loci across the genome that are repeatedly involved in parallel evolution (*Jones et al., 2012*). These loci are predominantly intergenic and thus may act through regulatory mechanisms.

During adaptation to their divergent environments marine and freshwater sticklebacks have evolved differences in numerous morphological, physiological and behavioral traits. Two key divergent traits include their anadromous (migratory marine) versus resident-freshwater life histories and the ability to live in fresh- and saltwater. For this adaptation, the gill's role in osmoregulation and respiration is likely to be particularly important (e.g. through regulation of ion channel genes, *Evans et al., 2005*). In saline water, fish counteract water loss and ion gain by ion exclusion. In freshwater, fish compensate against ion loss and water gain by ion uptake. Expression changes in osmoregulatory genes has been linked to freshwater adaptation by anadromous ancestors in sticklebacks and other fish (*Gibbons et al., 2017*; *Velotta et al., 2017*). Further, previous studies have shown that freshwater adaptation in sticklebacks and other fish is associated with changes in gene expression plasticity (*Gunter et al., 2017*; *McCairns and Bernatchez, 2010*; *Whitehead, 2010*). The genetic basis of these gene expression differences is not known but leads to the prediction that loci showing parallel divergence in gene expression levels show parallel and environmentally insensitive mechanisms of gene regulation in independently evolved marine and freshwater ecotypes.

Here, we study the evolution of gene expression and its *cis-* and *trans-* regulation in the gills of threespine sticklebacks as a model for regulatory evolution during early stages of parallel adaptive divergence with gene flow. Using freshwater-resident and anadromous marine sticklebacks from rivers in Scotland (3) and Canada (1), we ask to what extent parallel divergent adaptation to marine and freshwater environments involves parallel expression divergence in the gills under standardized laboratory conditions. We explore whether parallel differentially expressed genes are found more frequently near previously identified adaptive loci and whether they show molecular signatures of natural selection. We then dissect the *cis-* and *trans-* regulatory basis of gene expression differences using allele-specific expression analysis of marine-freshwater F1 hybrids and their parents. We ask whether *cis-* or *trans-* regulatory changes predominate in the early stages of adaptive divergence with gene flow, and, by comparisons across marine and freshwater ecotypes from four independently evolving river systems examine the degree of parallelism in *cis-* and *trans-* architecture.

Finally, by rearing F1 siblings in different water salinity conditions, we explore the extent to which the degree of *cis*- and *trans*-regulation of divergently expressed genes is influenced by the environment.

## Results

### Stickleback gill transcriptome assembly

We analyzed gene expression in the gill of marine and freshwater ecotype pairs each collected and derived from four river systems in Scotland and Canada (*Figure 1a*, *Supplementary file 1*). Gills of mature and reproductively active first-generation lab-raised female and male fish were dissected and their transcriptomes analyzed using strand-specific RNA-seq. We built a reference-guided assembly (*Trapnell et al., 2012*) of the stickleback gill transcriptome based on RNA-seq reads from 10 freshwater and 10 marine fish from 4 marine and four freshwater strains (*Supplementary file 2*). The stickleback gill transcriptome contains 29295 transcribed loci, 17304 of which are multi-transcript loci with 171620 different transcripts combined. This is considerably more than Ensembl gene build 90 for the stickleback genome (22456 loci, 29245 transcripts), likely because of the higher sequencing coverage of the current dataset compared to the low coverage EST libraries used to inform gene annotations of Ensembl's genebuild (*Kingsley et al., 2004*). The number of transcripts per locus is highly skewed with a median of 3 and a small number of loci, including genes with immune function, with very high numbers of splice forms (*Figure 1—figure supplement 1*). The transcriptome includes 7147 novel transcribed loci that do not overlap with any transcript in Ensembl gene build v90. Of these novel loci, candidate coding regions with complete open-reading frames and likelihood scores > 20 were identified for 1018 using TransDecoder (*Haas et al., 2013*). At the locus level our assembly shows very high sensitivity (Sn = 81%) with few false negatives (fSn = 100%) and moderate specificity reflective of the appreciable number of novel coding regions we detected relative to the Ensembl gene build 90 (Sp = 59%, fSp = 69%). Considering the raw FPKM data 21399 (73%) of loci were expressed at FPKM >= 1 in at least one of the 20 marine and freshwater fish analyzed, and 16195 (55%) in more than 10 individuals. Hierarchical clustering of expression levels revealed that the gill transcriptome can be characterized by five major groups of loci based on their average expression level (*Figure 1—figure supplement 2*) with the most highly expressed group of genes showing strong enrichment for biological processes with the respiratory function of the gill including mitochondrial respiration, ATP synthesis coupled proton transport and cytoplasmic translation (Appendix 1).

### Detecting parallelism in marine-freshwater transcriptome divergence in a largely non-parallel evolving transcriptome

We hypothesized that selection for divergent adaptation to marine and freshwater habitats could drive parallel divergence in gene expression in multiple independently evolving marine-freshwater ecotype pairs of distinct geographic origin. To test this hypothesis, we first investigated the major sources of covariation in the freshwater and marine transcriptomes using Principal Component Analysis (PCA).

While the first major axis of variation separates individuals by river system (24% variation explained, *Figure 1—figure supplement 3*), we identified PC2 and PC5 as major axes of variation (14.5% and 6.3% PVE, respectively, *Figure 1c*) that separate marine and freshwater transcriptomes. We defined a composite PC axis that captures parallel divergence in the transcriptomes of freshwater and marine ecotypes by summing the eigenvalue-weighted loadings from PC2 and PC5 (*Figure 1c*). The loadings of each transcript on this composite PC were used as a measure of each transcript's individual contribution to parallel transcriptome divergence. Parallel transcriptome divergence is highly correlated with river-specific measures of marine-freshwater transcriptome divergence (log of marine/freshwater expression fold change in the River Tyne and Little Campbell Rivers, *Figure 1—figure supplement 4*). Similar to the only 0.01% parallel genetic divergence observed at the genomic DNA level (*Jones et al., 2012*), here parallel divergence of gene expression between marine and freshwater ecotypes reared under common environmental conditions represents only a small proportion of transcriptome variation. From here on we refer to loci with the highest contribution toward parallel marine-freshwater expression divergence (top or bottom 1% composite PC

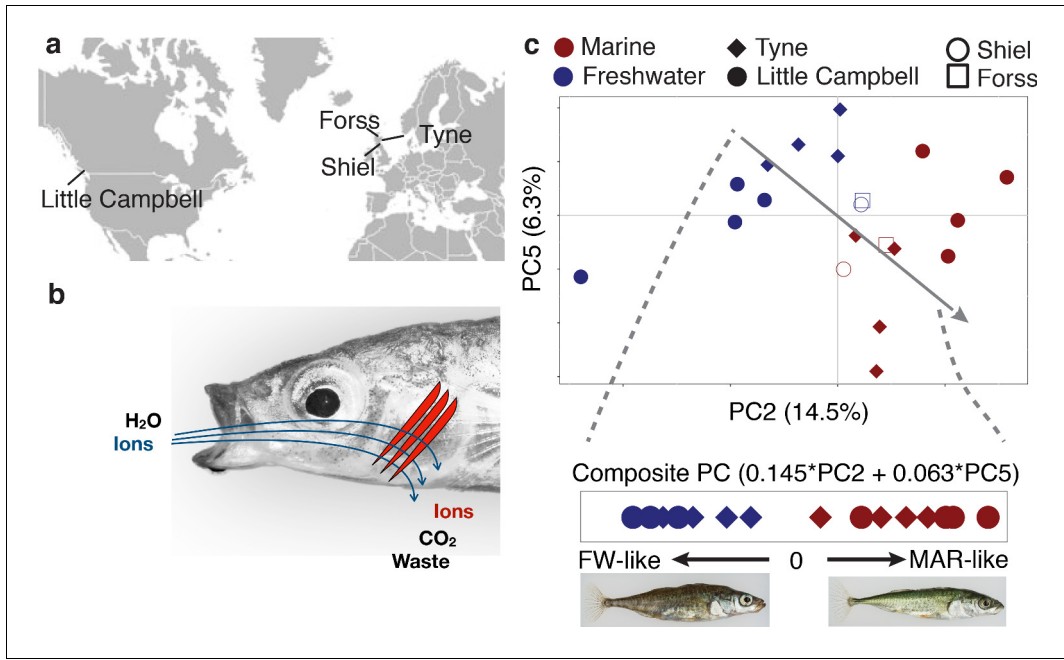

**Figure 1.** Freshwater and marine sticklebacks show parallel expression divergence among a largely non-parallel evolving transcriptome. (a) Marine and freshwater strains sampled from four different river systems (b) The gill is a multifunctional organ with roles in osmoregulation, respiration and waste excretion. In freshwater gills uptake ions (blue), while in saline water ions are excreted (red). (c) Principal components analysis of normalized expression levels separates marine (red) from freshwater (blue) ecotypes along a composite PC axis (gray line). PCA is calculated based on a sample-size balanced set of Tyne and Little Campbell samples (solid symbols), onto which Forss and Shiel individuals are projected (open symbols).

DOI: https://doi.org/10.7554/eLife.43785.002

The following figure supplements are available for figure 1:

**Figure supplement 1.** The stickleback gill transcriptome has a highly skewed distribution of isoform variation (number of 'TCONS') per expressed locus ('XLOC').

DOI: https://doi.org/10.7554/eLife.43785.003

**Figure supplement 2.** Hierarchical clustering of stickleback gill transcriptome by expression level in 20 marine and freshwater fish reared under standard laboratory conditions reveals five major groups of loci based on their average expression level.

DOI: https://doi.org/10.7554/eLife.43785.004

**Figure supplement 3.** Principal component analysis of gill transcriptomes of Little Campbell and Tyne ecotypes showing principal components 1 and 2 separating individuals by geography (PC1) and ecotype (PC2) respectively.

DOI: https://doi.org/10.7554/eLife.43785.005

**Figure supplement 4.** Parallel marine-freshwater divergence in gene expression quantified using the Composite PC loadings (y-axis, see also *Figure 1c*) is highly correlated with river-specific marine-freshwater divergence in gene expression River Tyne, Scotland (left) and Little Campbell River, Canada (right) quantified as the log of marine/freshwater expression values.

DOI: https://doi.org/10.7554/eLife.43785.006

**Figure supplement 5.** Assignment of parallel diverged genes and their FDR values.

DOI: https://doi.org/10.7554/eLife.43785.007

**Figure supplement 6.** Parallel diverged transcripts identified as per PCA analysis and parametric test overlap.

DOI: https://doi.org/10.7554/eLife.43785.008

loadings, median FDR over outliers 2.6%, N = 586 transcripts, *Figure 1—figure supplement 5*) as '*parallel diverged loci*'. Similar results were obtained from a differential expression analysis using a Cufflinks linear model (see Appendix 1, *Figure 1—figure supplement 6*).

The parallel diverged loci show an enrichment in gene ontology processes and molecular functions associated with gill ion exchange, osmoregulation and blood traits (*Supplementary file 3*). Among the overrepresented categories were multiple processes involved in ion transmembrane

transport, suggesting that the parallel diverged transcripts function in regulating osmolarity through ion exchange. Top ranking genes include Na-Cl cotransporter (slc12a10), Basolateral Na-K-Cl Symporter (slc12a2), cation proton antiporter 3 (slc9a3.2), Potassium Inwardly-Rectifying Voltage-Gated Channel (kcnj1a.3), potassium voltage-gated channel (KCNA2), Epithelial Calcium Channel 2 (trpv6), Sodium/Potassium-Transporting ATPase (atp1a1.4), aquaporin 3a (aqp3a), and carbonic anhydrase 2 (ca2) — genes known to play a role in osmoregulation in fish and other organisms. We also note that our analyses identified differentially expressed novel loci that have no overlap with gene annotations from the Ensembl gene build. These results are consistent with our hypothesis that adaptive expression divergence influences physiological functions of the gill associated with a transition to permanent freshwater environment.

## Natural selection on parallel expression divergence

To explore the role of natural selection in parallel expression divergence, we used adaptive loci identified in a previously published study (*Jones et al., 2012*) and analyzed newly generated whole genome sequence data of six unrelated fish of each ecotype from both the River Tyne and Little Campbell River for molecular signatures of selective sweeps.

Transcripts with parallel expression divergence are distributed across all 21 chromosomes (*Figure 2a*), and more proximal to genomic regions undergoing parallel marine-freshwater adaptive divergence at the DNA sequence level than expected by chance (identified in *Jones et al., 2012*), randomization test, $p<<0.025$; *Figure 2b–c*). More than 13% of transcripts with parallel expression divergence were found within 10 kb of regions of parallel genetic divergence; and nearly 40% within 75 kb (*Figure 2c*), consistent with the hypothesis that the predominantly intergenic marine-freshwater adaptive loci identified in *Jones et al. (2012)* contain regulatory elements contributing to parallel marine-freshwater divergence in expression. We found similar results when we calculated parallel genetic divergence (CSS) based on the whole genome sequences generated in this study (Appendix 1, *Figure 2—figure supplement 1*).

Natural selection is expected to reduce the diversity of a local genomic region leaving detectable molecular signatures of selection such as increased genetic divergence ($F_{ST}$) and reduced diversity (Pi) around adaptive loci (*Nielsen et al., 2005*). We calculated genetic divergence and nucleotide diversity flanking transcription start sites (TSSs). In Little Campbell fish, we observed a reduction in nucleotide diversity (Pi) within each ecotype around the TSSs of transcripts with parallel expression divergence (*Figure 2f–g*). Reduced within-population diversity was accompanied by increased genetic divergence ($F_{ST}$) that was centered on TSSs with a slight upstream bias (*Figure 2e*). In contrast, although we observed a slightly increased $F_{ST}$ around the TSSs of transcripts with parallel expression divergence in the River Tyne, we did not detect a reduction in nucleotide diversity relative to other transcripts (*Figure 2—figure supplement 2*), possibly due to concurrent selective sweeps on different haplotype backgrounds (also referred to as soft sweeps) in this younger population. Analysis of parallel differentially expressed transcripts defined using parametric tests gives similar results (Appendix 1, *Figure 2—figure supplement 3*). Combined, these results provide evidence for natural selection acting on parallel differentially expressed transcripts in the gill.

## Expression divergence between stickleback ecotypes is predominantly due to *cis*-regulatory changes

We hypothesized that the observed molecular signatures of selection around genes with divergent expression results from natural selection acting on *cis*-regulatory elements controlling gene expression levels. To investigate the role of *cis*-regulation, we compared the level of gene expression divergence in the gill transcriptomes of marine and freshwater parents to the level of allele-specific expression (ASE) in their reproductively mature F1 hybrid offspring using laboratory reared freshwater and marine strains from four independent river systems (Tyne, Forss, Shiel and Little Campbell).

Since F1 hybrids carry a marine and freshwater copy of each chromosome within a shared *trans* environment in each cell, any marine or freshwater allele-specific bias in transcript expression can be attributed to *cis*-acting regulatory elements, rather than *trans*-acting factors. We sequenced the parental genomes and identified coding-regions containing polymorphisms that were fully informative for allele-specific dissection (where the parents of a given cross were homozygous for alternate alleles, *Figure 3—figure supplement 1*, *Figure 3—figure supplement 2*). We classified regulatory

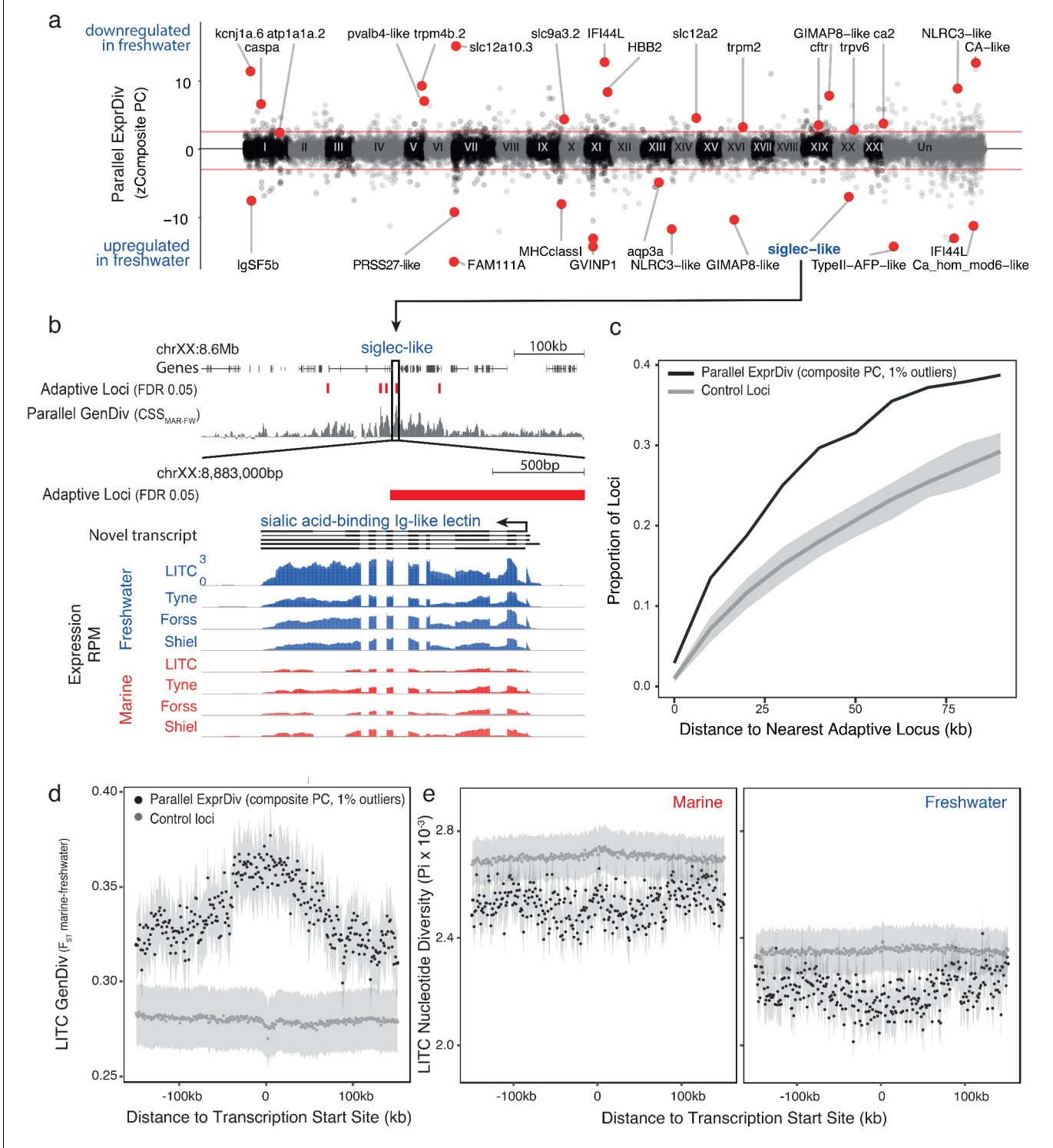

**Figure 2.** Genes with parallel marine-freshwater expression divergence are proximal to genomic regions underlying marine-freshwater adaptive divergence and show molecular signatures of natural selection. (**a**) Z-standardized composite PC loadings (zComposite PC) for each analyzed transcript across the genome highlighting transcripts with strong parallel marine-freshwater divergence in gene expression. Red lines correspond to the upper and lower 1% composite PC quantiles. Genes highlighted with red points have putative roles in ion transport and osmoregulation (*kcnj1a.6*, *atp1a1a.2*, *pvalb4*, *trpm4b.2*, *slc12a1*, *slc12a10.3*, *slc9a3.2*, *trpm2*, *trpv6*, *cftr*, *ca2*, *CA-like*, *aqp3a*), calcium homeostasis (*FAM111A*, *ca_hom_mod6-like*), respiration (*HBB2*), cold temperature adaptation (*TypeII AFP*), jaw, gill and skeletal morhpogenesis (*caspa*, *FAM111a*), and immune system function (*IFSF5b*, *caspa*, *PRSS27*, *MHCclassI*, *GVINP1*, *IFI44L* (*x2*), *GIMAP8-like* (*x2*), *NLRC3-like* (*x2*), *siglec*). (**b**) An example highlighting the proximity of parallel differentially

*Figure 2 continued on next page*

*Figure 2 continued*

expressed genes to adaptive loci is a novel transcript coding for a sialic-acid binding Ig-like lectin. This gene shows strong parallel expression divergence among freshwater and marine ecotypes from all four river-systems and overlaps a previously identified adaptive locus with a signature of parallel marine-freshwater genomic divergence (CSS FDR 0.05, *Jones et al., 2012*) (c) Across the genome, loci with strong parallel expression divergence (loci in the upper and lower 1% quantiles of composite PC, black line) are more proximal to adaptive loci identified in *Jones et al. (2012)* than randomly sampled 'control' loci (gray line). Gray shading shows 95% confidence intervals from 100 random samples of 586 transcripts. (**d and e**) In marine and freshwater fish from the Little Campbell River (LITC), loci with strong parallel expression divergence show molecular signatures of natural selection (d, increased genetic divergence $F_{ST}$; and e, decreased nucleotide diversity, Pi) centered around their transcription start sites (black points) compared to control loci (expressed loci showing non-parallel expression divergence). Points represent mean values of 1 kb sliding windows and gray shading shows the standard error of the mean.

DOI: https://doi.org/10.7554/eLife.43785.009

The following figure supplements are available for figure 2:

**Figure supplement 1.** Transcripts with parallel expression divergence are in close spatial proximity to regions of the genome that show elevated parallel genetic divergence (CSS outlier loci) among marine and freshwater fish from the River Tyne and Little Campbell that were genome sequenced in this study.

DOI: https://doi.org/10.7554/eLife.43785.010

**Figure supplement 2.** Genetic divergence ($F_{ST}$) calculated in 1 kb windows around the 586 loci with parallel divergent expression (black points with blue line representing mean) relative to randomly sampled loci (gray dashed line) in Tyne populations.

DOI: https://doi.org/10.7554/eLife.43785.011

**Figure supplement 3.** Genetic divergence (Fst) and nucleotide diversity (Pi) associated with loci showing parallel expression divergence defined based on a parametric test performed in cufflinks (Little Campbell).

DOI: https://doi.org/10.7554/eLife.43785.012

divergence into *cis*-acting (allele-specific), *trans*-acting (non allele-specific) categories, or a combination of the two, based on the allele-specificity and magnitude of expression divergence in F1 hybrids compared to marine and freshwater parents (*Figure 3a–c*, *Supplementary file 4*) (*Landry et al., 2005*; *Wittkopp et al., 2004*). Since we are interested in genetic variation in natural populations, we used first-generation wild-derived parents and analyzed four F1 offspring per parental pair, in order to account for the different genetic backgrounds of each offspring. Considering all expressed loci in the gill, 3807 (13%), 4472 (15%), 7716 (26%) and 10102 (34.5%) non-sex-biased genes could be assayed for *cis*- vs *trans*-mediated expression in at least one F1 individual from Tyne, Shiel, Forss and Little Campbell, respectively.

In each ecotype-pair, marine-freshwater expression differences are predominantly driven by divergence in *cis*-regulation (*Figure 3b*, *Figure 3—figure supplement 3*), with the degree of *cis*-divergence scaling positively with the degree of genetic divergence (measured either as the number of fixed differences or mean per site nucleotide divergence between the parents) between marine and freshwater ecotypes from each river system. Across the four river systems, from ~350 to ~2000 transcripts were classified as *cis*-diverged (with Little Campbell River fish showing the highest number,~2000, or ~20%, of *cis*- divergence. See also *Figure 3—figure supplement 4*).

## Loci with parallel expression divergence are enriched for *cis*-regulation

We next tested whether particular types of gene expression regulation were more likely to contribute to marine-freshwater expression divergence that has evolved in parallel in multiple river systems. We used 'parallel diverged loci' defined by the composite principal component described above that also show marine-freshwater expression differences in the same direction between the parents of our crosses and contain fully informative SNPs to allow for allele-specific analysis. After filtering, 181 and 68 parallel diverged loci were testable in Little Campbell and Tyne, respectively. Loci with parallel divergence in expression showed a significant excess of *cis*-regulation (16–26% and 11–12% above average in Tyne and Little Campbell F1 offspring respectively) and *cis+trans* regulation (above average), compared to random expectations obtained by 1000 random draws of equal size from all genes that showed expression divergence between ecotypes (see Materials and methods) (*Figure 4*). In contrast, the overrepresentation of *trans*-acting divergence was lower, between 6–16% and 0–2%, respectively, and not different from random expectation in all F1s.

In parallel diverging transcripts, ecotype-pairs from both rivers further showed a significant excess of *cis*- and *trans*-regulation of expression divergence acting in the same direction (divergence in *cis*

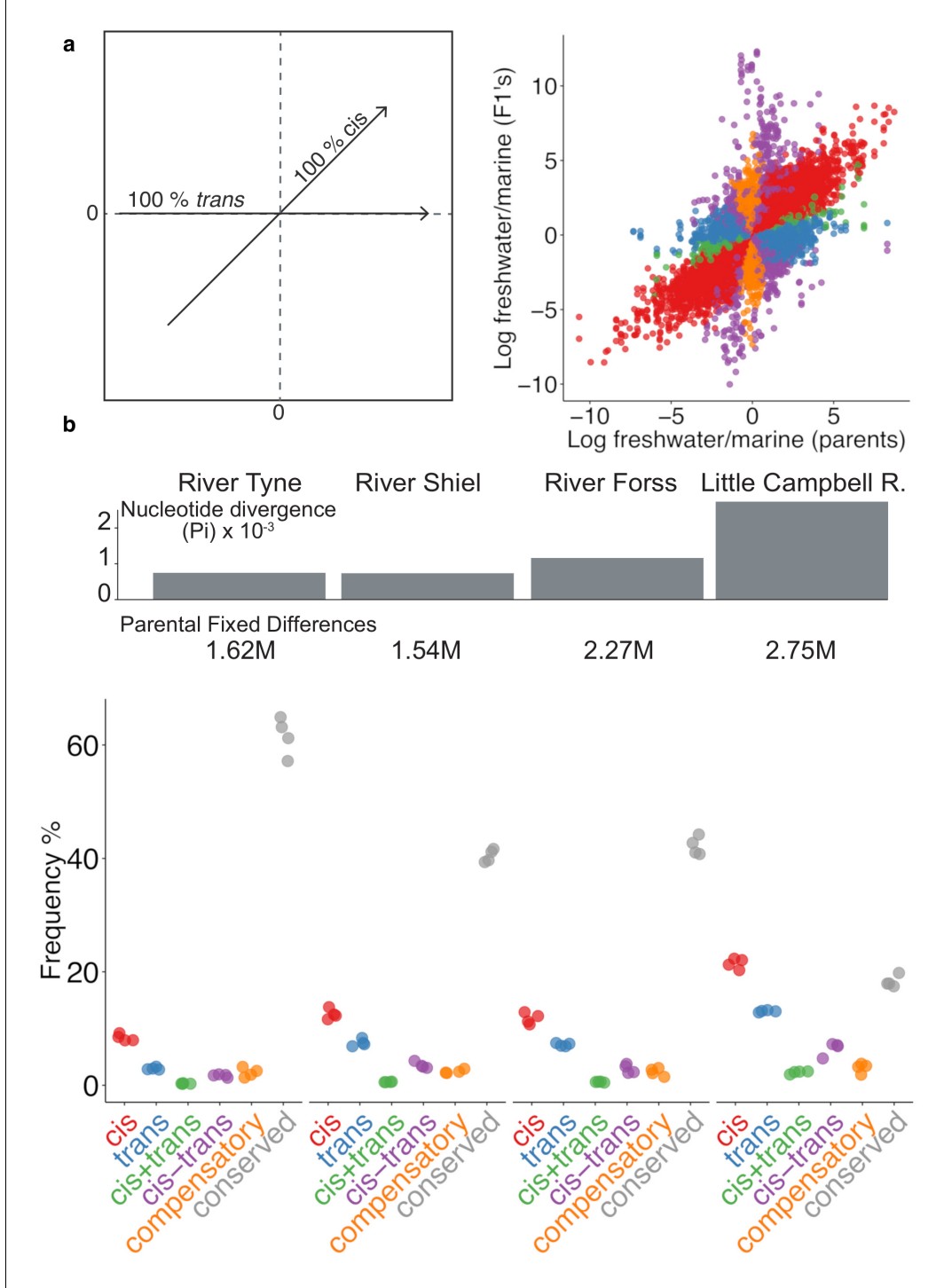

**Figure 3.** Expression divergence between marine-freshwater ecotype pairs is predominantly due to *cis*-regulatory divergence. (**a**) Regulatory divergence is categorized in six categories by comparison of the expression log₂-ratio of the marine and freshwater parent to marine- and freshwater- allele-specific expression log₂-ratio within an the F1 offspring (*Landry et al., 2005*). Individual data-points correspond to allele-specific expression values (y-axis) for each gene in each of four F1s relative to their parents (x-axis). Genes are colored according to their classification into different genetic architectures of expression divergence (red *cis*, blue *trans*, green *cis+trans*, purple *cis-trans*, gold *compensatory*, gray *conserved*. '*Ambiguous*' and '*conserved*' classes are omitted for clarity.) (**b**) *cis*-regulation is the predominant regulatory mechanism underlying gene expression divergence in all four river systems. Points show the overall frequency of regulatory divergence types, relative to the number of analyzed transcripts for each

*Figure 3 continued on next page*

*Figure 3 continued*

F1 offspring analysed. The proportion of expression divergence caused by *cis*-regulatory divergence, scales positively both with nucleotide divergence (gray bars) and the number of fixed differences (numbers beneath bars) between the marine and freshwater parents. '*Ambiguous*' class is omitted for clarity.

DOI: https://doi.org/10.7554/eLife.43785.013

The following figure supplements are available for figure 3:

**Figure supplement 1.** Number of informative (tested) SNPs per transcript per cross.

DOI: https://doi.org/10.7554/eLife.43785.014

**Figure supplement 2.** Expression levels of transcripts with informative SNPs.

DOI: https://doi.org/10.7554/eLife.43785.015

**Figure supplement 3.** The genetic architecture of expression divergence between ecotype pairs in all investigated ecotype-pairs (as per *Figure 3b*).

DOI: https://doi.org/10.7554/eLife.43785.016

**Figure supplement 4.** Frequencies of genetic architectures of expression divergence in down-sampled (30M uniquely mapping reads per sample) dataset.

DOI: https://doi.org/10.7554/eLife.43785.017

*+trans* co-regulation). Theory and experiments predict that directional selection on expression levels favors the accumulation of *cis*- and *trans*-regulatory changes that have the same direction of effect and act in a cooperative manner toward greater expression divergence (*Fraser et al., 2010*; *Orr, 1998*). The observed excess of *cis+trans* regulation in the transcripts that contribute to parallel expression divergence, is consistent with directional selection, as opposed to genetic drift, which should influence amplifying (*cis+trans*) and canceling (*cis-trans*) divergence equally (*Fraser et al., 2010*). Notably, the candidate genes slc12a2, atp1a1.2, trpv6 and aqp3a that are involved in teleost ion homeostasis were among the parallel diverged transcripts influenced by *cis+trans* acting regulatory divergence.

We next explored the extent of parallelism in the strength (magnitude) of *cis*- and *trans*-regulation among the four populations and stratified loci by their degree of contribution to parallel expression divergence (composite PC loading). For each F1 individual at each locus, we quantified *cis*-divergence following *Wittkopp et al. (2008)* as the F1 allele-specific expression ratio, and from there, quantified *trans*-divergence by subtracting the F1 allele-specific expression ratio from the parental expression ratio (*Wittkopp et al., 2008*). For each river system, we then averaged the *cis*-divergence and *trans*-divergence values for each locus across each of the four offspring to obtain a measure of the mean degree of *cis*- and *trans*-divergence per locus (see Appendix 1). We analyzed this data with a sliding cut-off, which results in loci with nested extremes of contribution to parallel marine-freshwater expression divergence as measured by the loadings on the composite principal component axis. Overall, we see high correlations between river systems in the degree of *cis*- and *trans*- divergence of loci with parallel expression divergence (left and right ends of composite PC axis). However, the pattern is not symmetrical. Parallel divergent loci that are upregulated in freshwater show strong positive correlations among populations in their quantitative extent of both *cis*- and *trans*-divergence (*Figure 5*). This correlation is lost in loci that do not contribute to parallel expression divergence (composite PC loadings close to zero) and is less consistent among parallel divergent loci downregulated in freshwater fish. We then estimated the mean effect size (magnitude) of *cis*- and *trans*- divergence for subsets of loci and represented them in bins along the parallel expression divergence (composite PC loading) axis. Again, we found that the magnitude of *cis*- and *trans*-divergence is not symmetrical: divergently expressed genes upregulated in freshwater fish show greater effect size in both *cis*- and *trans*- components than their freshwater downregulated counterparts (*Figure 5c*, *Figure 5—figure supplement 1*).

## *Cis*-regulatory divergence is additive and insensitive to genetic and environmental context

In order to evaluate the possible factors that may facilitate *cis*-divergence as a predominant source of parallel regulatory differences, we investigated: the mode of inheritance of different regulatory classes; the consistency of regulatory divergence in different genetic backgrounds; and the

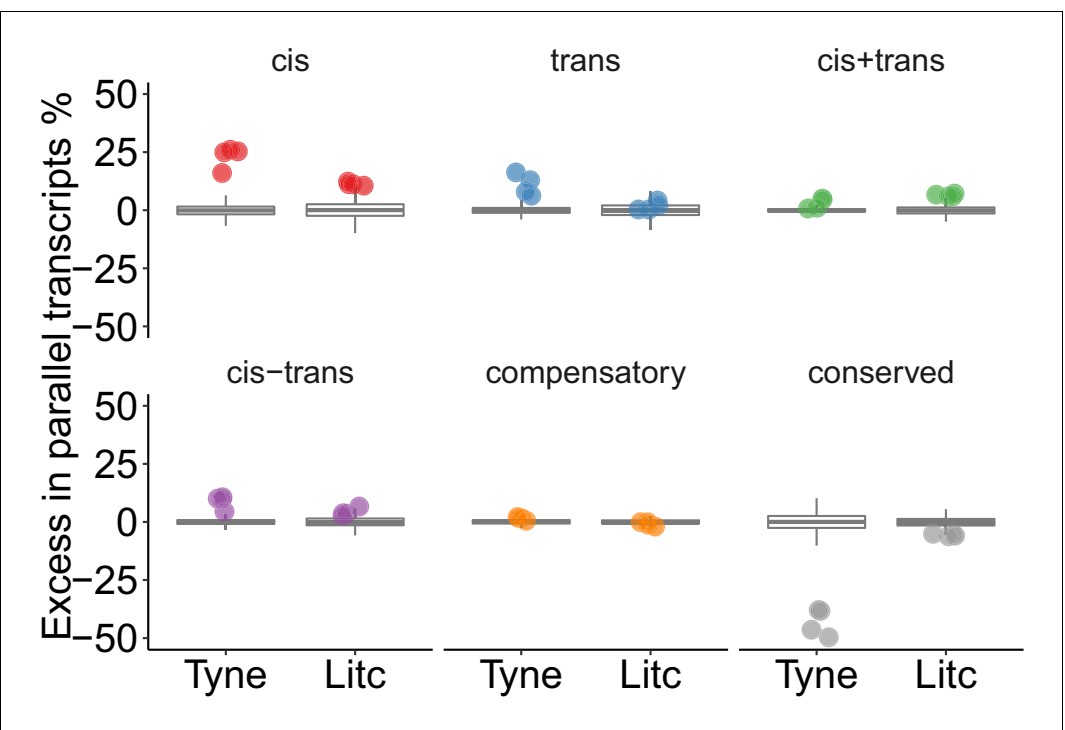

**Figure 4.** Genes with parallel marine-freshwater divergence in expression are enriched in cis-regulatory divergence. Overrepresentation of expression regulation types associated with parallel evolving transcripts, compared to the overall frequency of regulatory divergence between freshwater and marine ecotypes. Points represent the observed values in four F1's per cross. Box-plots represent the random expectation based on 1000 samples of the corresponding number from transcripts showing expression divergence between ecotypes. Horizontal bar in box-plot corresponds to median, box includes 50% and whiskers 99.3% of random expectation.
DOI: https://doi.org/10.7554/eLife.43785.018

consistency of regulatory divergence in different water salinities (a major environmental contrast between freshwater and marine ecosystems).

The mode of inheritance of a given trait, including expression traits, may have a direct impact on the rate of adaptation (*Lemos et al., 2008*). Among the modes of inheritance, additivity is associated with the greatest efficiency of bi-directional selection, because the contribution of each allele can be seen by selection (*Hartl and Clark, 1997*). To investigate this, we quantified the dominance/additivity ratio following *Gibson et al. (2004)* by taking the ratio between the mid-parental normalized read counts and the deviation from this mid-point in F1 hybrids. Taking into account all four F1-parent comparisons, *cis*-regulatory divergence showed the strongest level of additivity (*Figure 6a*), consistent with evolutionary potential for fast allele-frequency changes at *cis*-regulatory elements. These findings are consistent with previous studies showing that *cis*-divergence is linked to higher additivity of between-species expression differences (*Lemos et al., 2008*; *McManus et al., 2010*), except that here, we observed such divergence already among marine-freshwater ecotypes pairs with on-going gene flow, highlighting the role of *cis*-regulation in adaptive divergence even in nascent population pairs.

Adaptation is expected to favor non-epistatic alleles that confer a stable phenotype irrespective of the genetic background (*Stern and Orgogozo, 2009*). By using first-generation wild-derived parents in the crosses for this experiment we have maintained meaningful levels of genetic variation similar to those present in natural populations. The unique combination of parental genetic variation passed on to each F1 offspring enables us to explore the stability of gene expression regulation under different genetic backgrounds. We hypothesized that non-epistatic regulatory divergence should be consistent across siblings due to tight linkage, while epistatic regulatory divergence will leave a pattern of low correlation within the classes of regulatory divergence among F1 siblings. To quantify the level of epistasis in different classes of regulatory divergence we compared the

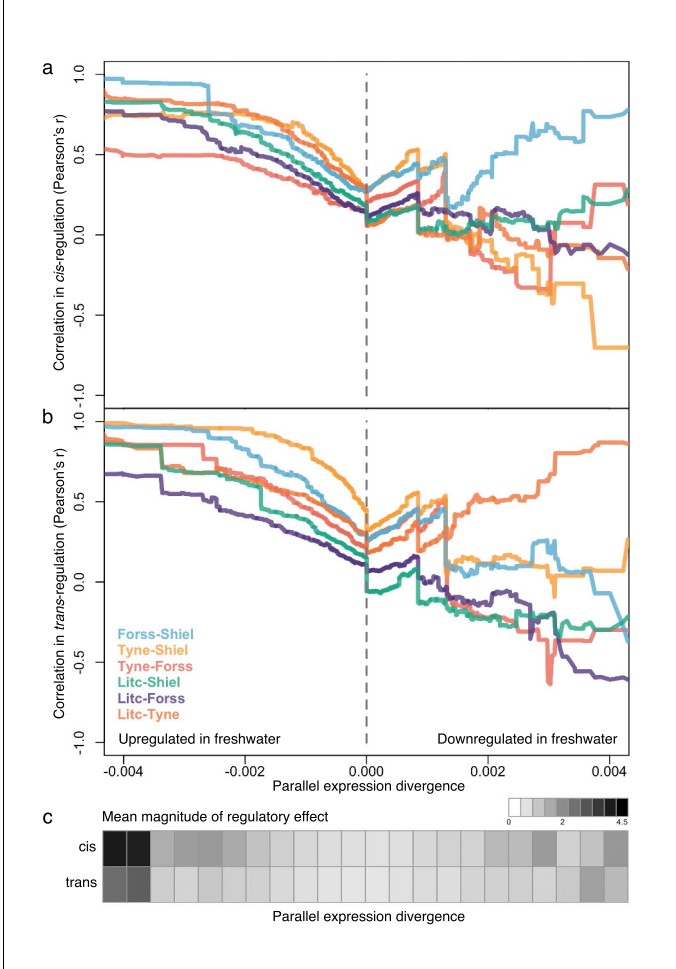

**Figure 5.** The degree of parallelism (correlation) in the magnitude of *cis*- (**a**) and *trans*-divergence (**b**) of marine-freshwater expression divergence among different river systems at loci with increasingly parallel divergent expression (composite PC loadings). Parallel divergent genes that are upregulated in freshwater (strong negative loading on composite PC) show high correlation in their degree of *cis*- and *trans*-divergence among populations. In contrast, loci that have not diverged in parallel (PC loadings close to zero) and loci that are parallel downregulated in freshwater are not highly correlated in the magnitude of *cis*- and *trans*- divergence. For each population pair, Pearson correlation coefficients, r, were calculated for subsets of loci defined by an increasingly extreme positive or negative threshold on the composite PC loading scores. (**c**) The mean absolute magnitude of *cis*- and *trans*- divergence across populations in subsets of loci defined by bins of parallel expression divergence (composite PC). For both *cis*- and *trans*-divergence, higher magnitude effects (darker gray shades) are seen at parallel divergent loci that are upregulated in freshwater. Means and standard errors of effect size per population are shown in *Figure 5—figure supplement 1*.

DOI: https://doi.org/10.7554/eLife.43785.019

The following figure supplement is available for figure 5:

**Figure supplement 1.** Mean magnitude of *cis*- (**a**) and *trans*- (**b**) regulation relative to parallel divergence in gene expression measured by composite PC.

DOI: https://doi.org/10.7554/eLife.43785.020

reproducibility of regulatory divergence across siblings of the same F1 family. Our results, consistent in both Tyne and Little Campbell crosses, show that *cis*-regulatory divergence tends to be most stable across genetic backgrounds (*Figure 6b* and *Figure 6—figure supplement 1*) making it a favorable substrate for adaptive divergence.

We next explored the stability in gene regulation under different environmental salinity conditions. Previous studies have reported changes in gene expression plasticity in association with

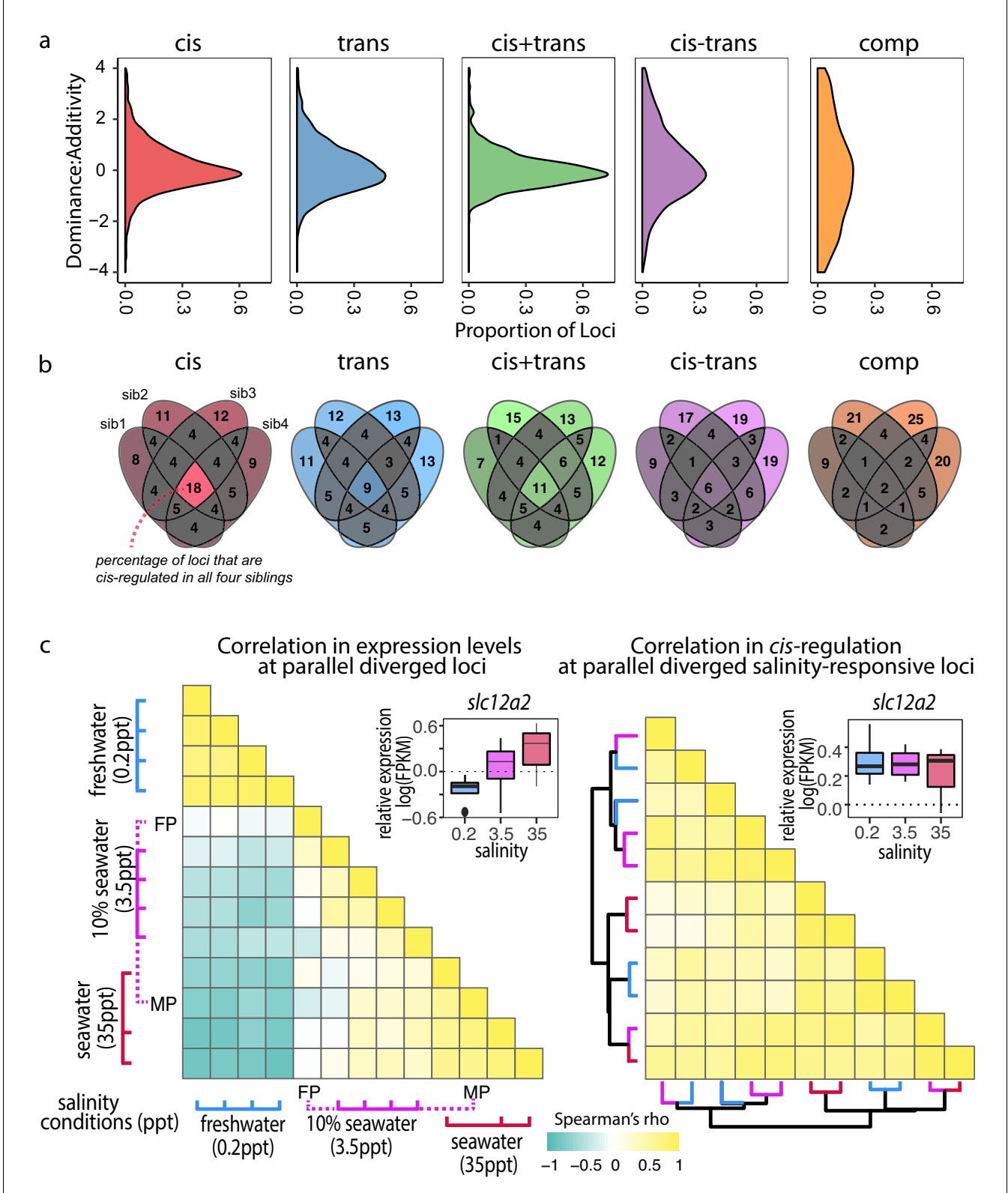

**Figure 6.** *Cis*-regulatory changes are additive and insensitive to genetic and environmental contexts compared to other regulatory classes. (a) Density estimates of dominance-additivity ratio of each locus assigned to the respective regulatory classes (data represents all four F1 – parent comparisons of Little Campbell). A ratio of zero indicates additivity, ±1 full dominance, and values > 1 or <-1 imply over- or underdominance. The slight bias towards negative values indicates tendency of stronger dominance of low-expressed allele (irrespective of ecotype). (b) Sharing of regulatory divergence

*Figure 6 continued on next page*

*Figure 6 continued*

between siblings as a proxy for the level of epistasis associated with each regulatory class in Little Campbell. The type of regulation observed for each gene is compared among siblings (overlapping ovals) with numbers representing percent of loci shared between siblings (rounded to integer). Color scale ranges from gray (low) to colored (high). *Cis*-regulatory divergence tended to be most stable across genetic backgrounds indicating that *cis*-acting divergence is least influenced by epistasis. (see also data for River Tyne shown in supplement figure). (c) While the expression levels of parallel diverged transcripts are sensitive to water salinity (left triangle) the degree of *cis*-regulation of these loci is insensitive to water salinity (right triangle). Pairwise Spearman's correlation (*rho*) in expression levels (left triangle) and allele-specific-expression ratio (right triangle) of parallel diverged loci in marine x freshwater F1 siblings acclimated to different salinity conditions. A negative correlation (cyand) indicates salinity-responsive genes were up-regulated (or show allele-specific-expression bias) in one salinity and down-regulated (or show allele-specific-expression bias in the opposite direction) in another, while a positive correlation (yellow) indicates that expression profiles and allele-specific expression profiles were similar between individuals. Expression levels of an example locus, slc12a2, are shown as an inset above each heatmap. slc12a2 expression increases under seawater conditions, but *cis*-regulatory control of this expression is stable across different salinities. Rows and columns of the right heatmap are ordered based on euclidean distance and the dendogram shows that *cis*-regulation does not cluster by salinity. Colors of dendogram branches refer to salinity in parts per thousand (0.2-blue, 3.5-pink, 35ppt-red). MP, FP refer to marine and freshwater parents.

DOI: https://doi.org/10.7554/eLife.43785.021

The following figure supplements are available for figure 6:

**Figure supplement 1.** Sharing of regulatory divergence between siblings as a proxy for the level of epistasis associated with different genetic architectures in Little Campbell (see also main text *Figure 6b*) and Tyne.

DOI: https://doi.org/10.7554/eLife.43785.022

**Figure supplement 2.** Principal Component Analysis of expression profiles of full siblings from a Little Campbell River marine x freshwater cross acclimated to different salinities.

DOI: https://doi.org/10.7554/eLife.43785.023

freshwater adaptation (*Gibbons et al., 2017*; *Velotta et al., 2017*). We analyzed adult gill transcriptomic and *cis*-regulatory responses to water salinity using full-sib F1 hybrids raised under three different salinity conditions (0.2 ppt (freshwater), 3.5 ppt (approximately equivalent to 10% sea water) and 35 ppt (marine salinity, see Materials and methods for more information).

A total of 2542 transcripts were differentially expressed between at least two salinity treatments (FDR 1%), consistent with major changes in gill structure and function upon salinity acclimation (*Hwang et al., 2011*). Differences between salinity treatments were mostly driven by freshwater acclimation (0.2 ppt), which elicited a response markedly different from both standard husbandry conditions (3.5 ppt) and typical marine salinity (35 ppt) (*Figure 6c*, *Figure 6—figure supplement 2*). We observed that an unexpectedly high proportion of parallel diverged transcripts also showed an effect of salinity on expression level (97 of 586, hypergeometric test p<4.3e-6), suggesting a large proportion of the parallel diverged transcripts in the gill are involved in physiological responses to water salinity.

Plasticity in expression of a given genotype in different environments can be mediated by either *cis* or *trans*-regulatory mechanisms. Focusing on the 97 transcripts with parallel expression divergence that showed a salinity response, we asked whether the regulatory control of these loci is sensitive to salinity conditions. We first calculated an expression profile for each F1 sibling by scaling the FPKM expression of each transcript to the average expression level across all individuals and then compared the profiles using Spearman correlation to capture the strength and direction of correlation in expression profiles among salinity treatments. F1's raised in similar salinities tended to have high positive correlation coefficients while the expression profiles of individuals raised in freshwater tended to be negatively correlated (opposite expression) with expression in individuals raised in salt (*Figure 6c*, left triangle).

Then, using allele-specific expression as a proxy for *cis*-regulatory divergence we investigated the stability of *cis*-regulation of expression across the salinity treatments. Despite the observed plastic response of gene expression (*Figure 6c*, left triangle), the degree of *cis*-divergence of salinity-responsive genes with parallel expression divergence was highly correlated across salinities (*Figure 6c*, right triangle), indicating that the observed salinity response was not caused by *cis*-regulatory divergence but largely due to *trans*-acting regulation that influenced the expression of both alleles in similar magnitude. Based on these results we hypothesize that *cis*-regulatory changes provide a mechanism for genetic assimilation of plastic responses into heritable variation (*Waddington, 1942*) where the effects of regulatory alleles are independent of the environment.

## Discussion

Regulation of gene expression is thought to play a major role in adaptation, yet relatively little is known about the patterns and predictability of adaptive regulatory mechanisms in the early stages of intra-specific adaptive divergence that evolve in the face of on going gene-flow. We characterized parallel expression divergence in the threespine stickleback gill transcriptome - an organ with important respiratory and osmoregulatory functions - and dissected the *cis*- and *trans*- regulatory architecture in four marine-freshwater ecotype pairs in order to infer the rules and patterns shaping genome evolution and influencing rapid adaptation in natural populations.

We found that when sticklebacks are reared under the same standard husbandry conditions, parallel transcriptomic divergence involves only a few hundred genes each with relatively small effect on expression divergence, similar to the small proportion of the genome that shows parallel adaptive divergence at the DNA sequence level (*Jones et al., 2012*). Parallel divergently expressed genes are proximal to adaptive loci identified from signatures of parallel divergence at the sequence level, and support a role for the reuse of ancient standing genetic variation in the parallel adaptive divergence of gene expression. We also observed a large proportion of loci with marine-freshwater divergent expression that was unique to a local river system (see Appendix 1) indicating there is room for drift and/or local adaptation to play a major role shaping the evolution of divergent gene regulation and gene expression in any given river system. The transcripts showing strongest parallel expression divergence are associated with genes involved in gill ion transport and osmoregulation, a key physiological trait in marine-freshwater divergence, and in agreement with studies in other fish (*Velotta et al., 2017*). Through population genomic analysis, we show how natural selection on divergently expressed genes is shaping the evolution of the genome - strong molecular signatures of selection (elevated $F_{ST}$, reduced Pi) were detected around the transcription start sites of parallel divergently expressed genes.

We see overwhelming evidence for the importance of *cis*-regulatory divergence underlying marine-freshwater gene expression divergence through analysis of allele-specific expression in F1 hybrids relative to their marine and freshwater parents. The predominance of *cis*-divergence was observed in marine-freshwater ecotypes from four independent river systems, in analysis of all informative transcripts genome wide, and enriched in the set of loci identified as having evolved parallel marine-freshwater divergence in expression across rivers.

Many studies have reported that *cis*-regulatory differences contribute strongly to divergence in gene regulation over long evolutionary divergence scales (*Coolon et al., 2014*; *Emerson et al., 2010*; *He et al., 2016*; *Lemmon et al., 2014*; *Lemos et al., 2008*; *Prud'homme et al., 2007*; *Stern and Orgogozo, 2009*). The stickleback ecotype pairs studied here diverged from each other within the last 10–20 k years (approximately 10–20 k generations) following the retreat of the Pleistocene ice sheet, indicating that strong *cis*-regulatory divergence can also evolve in relatively short timescales. Instead of fixed rules governing the major modes of regulatory divergence, the genetic basis of adaptive regulatory changes will likely depend of the genetic architecture of the trait under selection. For example during maize domestication, while *cis*-regulatory divergence did not dominate, it did contribute to stronger magnitude regulatory changes over a relatively short evolutionary time-span (*Lemmon et al., 2014*). It should also be noted that while we observe an association between nascent ecotype divergence and *cis*-regulatory divergence, within population *cis*-regulatory variation can be initially produced through non-adaptive processes, for example relaxed purifying selection (*Steige et al., 2017*). While adaptation via re-use of ancient standing genetic variation is important in sticklebacks and may explain some part of the predominance of *cis*-regulatory changes, we also found that the extent of genomic divergence varied substantially between the parallel evolving stickleback ecotype-pairs we studied, and the proportion of *cis*-regulatory divergence scaled positively with this genomic divergence. This suggests that much of the genetic differences that accumulate in the early stages of adaptive divergence with gene flow translate to *cis*-regulatory differences. Mutation-accumulation experiments have shown that genetic drift, which promotes random regulatory changes, is biased toward *trans*-acting divergence due to a larger *trans*-mutational target size (*Landry et al., 2007b*). Since most expression changes between freshwater and marine ecotypes, however, tended to be due to divergence in *cis*-, this points toward a stronger contribution of selection rather than drift.

Our results further indicate that reinforcing *cis* and *trans*-acting regulatory divergence that act in a manner to amplify one another, although rare, has an important contribution to early ecotype divergence with gene flow. Overrepresentation in *cis+trans* coevolution seemed to grow with increasing genetic and expression divergence between lineages, which is notable in the context of incipient ecological speciation. Given enough evolutionary time, the accumulation of amplifying *cis +trans* regulatory divergence independently in diverging populations may lead to the evolution of genetic (Bateson-Dobzhansky-Muller) incompatibilities. Because recombination between coevolved *cis*- and *trans*-regulatory factors disrupts their combined phenotypic effects, it is expected that selection would favor linkage disequilibrium between the coevolved regulatory factors (*Verta et al., 2016*). Association between increasing proportion of divergence in *cis+trans* regulation and genome-wide divergence as seen here suggests that selection against recombination between coevolved regulatory factors may contribute to increased genetic divergence as adaptation proceeds and thus shape the genomic landscape of incipient speciation.

Parallel adaptive divergence of marine and freshwater ecotypes provides biological replicates of the evolutionary process which we can interrogate to identify common patterns governing the molecular basis of adaptation. We not only identified parallelism across marine-freshwater ecotype pairs in the predominance of *cis*- divergent gene expression, but also parallelism in the quantitative extent of both *cis*- and *trans*- regulation of divergently expressed loci. This strong parallelism was particularly notable in parallel divergently expressed genes that are upregulated in freshwater ecotypes, and considerably less strong and less predictable in parallel divergently expressed genes that are downregulated in freshwater. Similar to the underlying shared genetic basis of freshwater adaptation due to the parallel reuse of standing genetic variation, our results suggest we can not only predict that freshwater populations are likely to carry the same alleles at adaptive loci across the genome, but also that the magnitude and extent of *cis*- and *trans*- upregulation of divergently expressed genes is likely to be shared among freshwater populations. It is possible that there are fewer molecular mechanisms available to freshwater fish to 'turn on' gene expression and more diverse mechanisms to 'turn-off' gene expression. Coinciding with the stronger parallelism in genes upregulated in freshwater, we observed a tendency for the *cis*- and *trans*-regulatory effect size to be of larger magnitude than the *cis*- and *trans*-regulatory effects of genes downregulated in freshwater. It is plausible that this larger effect size enables selection to be more efficient and contributes to the observed stronger parallelism. Further, the marine ecotype is thought to be the ancestral state in threespine sticklebacks and has been evolving in a comparatively stable marine environment for millions of years. Under this evolutionary context, freshwater populations have a comparatively smaller effective population size with reduced access to pre-existing adaptive standing genetic variation, and have been subject to more recent and potentially stronger selection pressures than their marine ecotype counterparts. Thus, marine ecotypes have more opportunity to evolve via soft sweeps on standing genetic variation. Combined this evolutionary context is likely to have influenced which molecular mechanisms of gene expression regulation are visible, and efficiently respond to selection, resulting in relatively few paths to evolve adaptive upregulation of gene expression in freshwater populations compared to marine.

We note that our results contrast with a recent study by *Hart et al. (2018)* who reported a predominance of, and parallelism in, the *trans*-regulatory control of marine-freshwater divergent gene expression in the pharyngeal tooth plate. One possible biological explanation for this might be differences in the multifunctional (pleiotropic) functional roles of the gill and its likely complex genetic architecture compared to dental tissue with a less pleiotropic functional role and more simple genetic architecture involving a few large effect loci upstream of other factors (*Cleves et al., 2014*) that cause a dominant *trans*-regulatory signal in the allele-specific expression assay.

Our results indicate that parallel evolving divergence may converge on *cis*-regulatory changes driven in part by a higher level of additivity and lower level of epistasis of *cis*-regulatory factors. Known as the effect of '*Haldane's sieve*', the fixation of beneficial recessive alleles are hindered due to its phenotype being hidden from selection (*Haldane, 1927*). When alternative alleles are favored in their respective environments, as seem to be the case in sticklebacks, additive alleles have the highest likelihood of becoming fixed in both populations as both alleles can rise quickly to a high frequency. *Cis*-regulatory variation also tended to have a lower level of epistasis within populations compared to other types of regulatory effects. Epistatic regulatory alleles tend to have different phenotypic effects depending on the genetic background, therefore inducing unpredictable fluctuation

in expression levels. Hence, parallel evolution of gene expression seems to favor non-epistatic regulatory alleles that have similar effects on expression levels independently of the genetic background.

In addition to promoting divergence in expression levels that is independent of genetic backgrounds, adaptation-with-gene-flow is predicted to promote reduced expression plasticity, especially in cases where selection is strong (*Stern and Orgogozo, 2009*). Toward this end, our results indicate that divergence in *cis*-regulation may play a particularly important role in the evolution of reduced plasticity. Since plasticity is known to be involved in the divergent adaptation of marine and freshwater sticklebacks we investigated the sensitivity of gene expression and its *cis*- and *trans*-regulation to environmental salinity. Genetic assimilation involves the heritable encoding and loss of plasticity of a once plastic trait (*Waddington, 1942*). Previous studies have found evidence for genetic assimilation in gill gene expression evolution between stickleback ecotypes (*McCairns and Bernatchez, 2010*). While we found a strong component of plasticity in gene expression among siblings raised in different salinities, the *cis*-regulation of this gene expression was stable and insensitive to differences in the environmental conditions. From this we infer that the observed plasticity in expression is likely mediated via *trans*-regulation, and hypothesize that the stable divergence in *cis*-regulation component may serve as a mechanism for genetic assimilation.

The importance of *cis*-regulatory divergence underlying parallel marine-freshwater expression differences has implications for our understanding of genome function in natural populations in the early stages of adaptive divergence. The loci with parallel expression divergence that are predominantly regulated by *cis*-acting changes are dispersed throughout the genome but proximal to previously identified adaptive loci. We observed signatures of selection around their transcription start sites likely to be a result of natural selection acting on *cis*-regulatory elements. We show that *cis*-regulatory divergence acts in an additive manner and is robust to both different environmental contexts, and potential epistasis caused by differences in genetic background. Comparing across marine-freshwater ecotype pairs from independent river systems, we observed parallelism and large magnitude effect sizes for *cis*-regulation of divergently expressed loci upregulated in freshwater. These features make *cis*-divergence a well-poised target for natural selection and may explain parallelism in the predominance and quantitative extent of *cis*-divergence in the early stages intraspecific adaptive divergence. Combined our study highlights how natural selection on adaptive *cis*-divergence is a likely contributor to the dispersed genomic landscape of adaptation in sticklebacks.

## Data access

Data has been deposited to the Sequence Read Archive under the accession PRJNA530695. All scripts used in data analysis are available at https://github.com/jpverta/verta_jones_elife_2019.git (*Verta, 2019*; copy archived at https://github.com/elifesciences-publications/verta_jones_elife_2019).

## Materials and methods

### Establishing common garden strains

We captured freshwater-resident and anadromous marine sticklebacks from Little Campbell River, Canada, and Rivers Tyne, Forss and Shiel, Scotland, with wire-mesh minnow traps. Sampling locations are given in *Supplementary file 1*. We identified freshwater resident and anadromous marine ecotypes based on their lateral plates. Most freshwater-resident populations of sticklebacks are low-plated, whereas anadromous marine forms exhibit complete lateral plating (*Bell and Foster, 1994*). Consistently, in most cases the large majority of fish captured in freshwater were low-plated and anadromous marine fish captured near the mouth of the river/lake were fully plated. We generated within-ecotype crosses via in vitro fertilization of gravid females with males within ecotypes and transported the fertilized eggs to a common-garden environment at the Max-Planck campus in Tübingen in reverse-osmosis water supplemented with Instant Ocean salt to 3.5 ppt (~10% sea water salinity). No significant mortality occurred during transfer. The Max Planck Society holds neccessary permits to capture and raise sticklebacks. All animal experiments were done in accordance to EU and state legislation and avoiding unneccessary harm to animals.

## Fish used for the study

The fish used as 'pure ecotypes' and 'parents' were lab-reared but direct descendants of wild-caught individuals with the exception of one Tyne freshwater male individual, which was descendant of multi-generation lab-raised individuals. The first-generation lab-grown fish were raised in the lab from egg to adult and then crossed to produce F1's. In our study, we refer to 'parents' as the actual parents of the 'F1's' analysed.

Compared to plants, transgenerational inheritance of epigenetic changes is minimal in vertebrates, and we attribute expression differences observed in the 'pure ecotype'/'parent' as well as the 'F1' generations to genetic differences. While we cannot rule out the possible contribution of transgenerational plasticity (e.g. maternal effects from wild caught fish) contributing to expression differences among the lab-reared 'parents', we note that this would not influence allele specificity of F1 hybrid offspring expression. If anything, our results are therefore likely to be conservative underestimates of the role of divergent *cis*-regulation of expression. Further, we argue that compared to organisms such as placental mammals with prolonged *in utero* development, the influence of transgenerational plasticity on gene expression of externally fertilized and reared sticklebacks is likely to be much weaker.

The fish were raised in individual 100 liter tanks with 40–50 fish per tank in 3.5 ppt salinity and alternating light cycle of 16 hr light and 8 hr darkness of 6 months. No significant mortality occurred during captivity. Fish were raised with a diet of fry (freshly hatched artemia), juvenile (artemia, Daphnia and cyclops) and adult food (bloodworm, white mosquito larvae, artemia, mysis shrimp and Daphnia) until adults. We selected four fish from independent field crosses per ecotype from Little Campbell River and River Tyne strains, and one fish per ecotype from Shiel and Forss strains. Exception to this was one Tyne freshwater male fish that was the progeny of unrelated lab-raised freshwater parents, which was included to complete the sampling. We in-vitro crossed one marine female and one freshwater male for each strain, and raised the F1 individuals in identical conditions as the parents until they were reproductively mature (each cross in individual tank).

For the Tyne cross, we separated the F1 clutch into three at 3 months of age and transferred the F1s into separate 100 liter tanks with 3.5 ppt water. We added either 0.2 ppt or 35 ppt water in increments of 20 l at a time twice a week over the course of 1 month to acclimate fish in two of the tanks to different water salinities. After 1 month of acclimation all water in the two tanks was changed to either 0.2 or 35 ppt. We raised the fish in 0.2, 3.5 or 35 ppt water for additional three months until reproductive maturation.

## Sample preparation and RNA-sequencing

We harvested gill tissue for all strains and F1s, all staged as adults and reproductively active (gravid females and males exhibiting mating coloring). We flash-froze gills on liquid nitrogen and stored in −80˚C until used for mRNA-extraction. We disrupted gill tissue with a pestle on liquid nitrogen and extracted mRNA using Dynabeads mRNA direct kit (Invitrogen) and following manufacturer's instructions, followed by DNase treatment with the Turbo DNA-free kit (Ambion). We verified mRNA quality using Agilent BioAnalyzer.

We used 150 ng of mRNA to construct strand-specific RNA-seq libraries using the TruSeq Stranded RNA-seq kit (Illumina). We verified library yield using Qubit and size distribution using Bio-Analyzer. We optimized library construction protocol to result in mRNA insert size distribution centered on 290 base pairs. We pooled the libraries in equimolar amounts and sequenced in pools of eight samples on a HiSeq-3000 instrument, producing 150 bp paired-end reads (*Supplementary file 2*). We included replicate sequencing libraries in different lanes of the same run and different runs of the same instrument in order to measure the effect of batch on final data (none observed).

## Gill transcriptome assembly from RNA sequencing

We verified read quality with *FastQC* software (http://www.bioinformatics.babraham.ac.uk/projects/fastqc/) and trimmed the reads of sequencing adapters using TrimGalore (http://www.bioinformatics.babraham.ac.uk/projects/trim_galore/).

We aligned RNA-seq reads from pure strains to the UCSC stickleback genome reference ('gasAcu1') with *STAR* aligner (*Dobin et al., 2013*). We opted for running *STAR* in two-pass mode, gathering novel splice junctions from all pure-strain samples for the second alignment pass. After

experimenting with alternative parameters, we opted for the following: *–outFilterIntronMotifs RemoveNoncanonicalUnannotated –chimSegmentMin* 50 *–alignSJDBoverhangMin* 1 *–alignIntronMin* 20 *–alignIntronMax* 200000 *–alignMatesGapMax* 200000 *–limitSjdbInsertNsj* 2000000.

We followed the *Cufflinks2.2* pipeline (*Trapnell et al., 2012*) for reference-guided transcriptome assembly and transcript and isoform expression level testing. We used *Cufflinks2* to assemble aligned RNA-seq reads into transcripts, using Ensembl gene models for stickleback (version 90) as guide and the following parameters: *–frag-bias-correct* gasAcu1.fa *–multi-read-correct –min-isoform-fraction* 0.15 *–min-frags-per-transfrag* 20 *–max-multiread-fraction* 0.5. We produced a single merged transcriptome assembly based on all pure strains using *CuffMerge* and used this in all subsequent analyses for all samples.

Finally, we used *CuffDiff* to test expression differences between male and female fish from freshwater and marine ecotypes of Tyne and Little Campbell river (combined). Transcripts with sex-dependent expression at FDR 10% (N = 278) were excluded from analysis of genetic architecture of expression divergence (see below), but included in all other analyses where the number of male and female fish were balanced across the experimental contrast.

## Principal component analysis

We summarized read counts over transcript models using the *Cufflinks* function *CuffQuant with 'fr-firststrand' strand-specific RNAseq library type and other settings as default* and normalized read counts to total library sizes using *CuffNorm*. We subsequently transformed the read data with the *DESeq2* (*Love et al., 2014*) function *varianceStabilizingTransformation* so that the variance in read counts was independent of the mean, following the steps outlined in the *DESeq2* manual. We used the *R* (*Ihaka and Gentleman, 1996*) function *prcomp* with the option *scale = FALSE* to calculate PCA on expression level co-variances using data from all transcripts. We calculated the PCA based on a balanced set of four freshwater and four marine ecotypes from both Tyne and Little Campbell, and projected the single ecotypes from Forss and Shiel to principal components 2 and 5 using the *R* function *scale*. We verified the absence of batch effects in the RNA-seq data by PCA analysis of a replicate sequencing library sequenced on different lanes of the same run and on different runs. Technical variation was much smaller than biological variation.

As described in the main text, a combination of principal components 2 and 5 best described freshwater-marine divergence in transcriptomes in our dataset. We therefore defined a composite principal component by summing principal components 2 and 5, weighing each with the percentage of variation explained. Finally, we extracted principal component 2 and 5 loadings for each transcript and used the identical approach to calculate transcript loadings on the composite principal component. This procedure produced a loading value for each transcript that described the importance of that transcript in parallel freshwater-marine expression divergence.

We calculated a False Discovery Rate (FDR) for the detection of loci with parallel expression divergence. Since no standard way exists to test the significance of single PCA loadings in the context of the overall PCA, we used the following approach inspired by *Linting et al. (2011)*.

The procedure has the following steps: (i) for each outlier gene shuffle the sample labels while keeping all other data as original, (ii) re-calculate PCA and composite PC, (iii) store the composite PC loading from the permutation, (iv) shuffle through all possible combinations of sampling two groups of 8 individuals from 16 and perform steps i-iii for each composite PC outlier to produce a null distribution of composite PC loadings, (v) calculate Z-score and p-value for observing a loading higher than the permuted null and (vi) correct for multiple testing (FDR).

The procedure tests the null hypothesis that the composite PC loading of a given gene is not significantly higher compared to composite PC loadings calculated from non-structure data (i.e. where the sample identities have been shuffled). The approach produces an FDR for each tested gene, in our case composite PC outlier at 1% threshold. We report the median FDR over all composite PC outliers (2.6%) and mean (5.1%).

## Gene ontology analysis

Tests for enrichment of genes involved in specific biological processes, molecular functions and cellular components among top ranking differentially expressed genes was performed using GOrilla (*Eden et al., 2009*). Genes were sorted by CuffDiff differential expression q-values or by composite

PC loading score and, because stickleback genes are largely unannotated for gene ontologies, were mapped to mouse orthologs (the vertebrate with the highest GO annotation quality (ref: https://www.ncbi.nlm.nih.gov/pmc/articles/PMC2241866/) using a REST command to access the Ensembl precomputed ortholog database. We tested for significant enrichment of gene ontologies (biological processes, molecular functions, and cellular compartments) with p-values less than $1 \times 10^{-5}$ (*Supplementary file 3*). A similar approach using both human and zebrafish orthologs revealed enrichment in many of the same gene ontologies (not shown).

## DNA preparation and sequencing

We extracted genomic DNA from fin clips of six unrelated fish per ecotype (six females of each ecotype from Little Campbell, three males and three females of each ecotype from Tyne) using standard lysis buffer and proteinase-K digestion, followed by SPRI bead extraction with Ambion magnetic beads. We verified DNA quality on agarose gel and quantified DNA concentration using Qubit.

We fragmented 700 ng of gDNA with Covaris instrument and selected 300–500 bp DNA fragments for library construction using double-sided SPRI selection. We constructed DNA-sequencing libraries using a custom protocol that includes DNA end-repair, A-tailing and Illumina TruSeq adapter ligation, followed by six cycles of PCR amplification. We verified library fragment size using BioAnalyzer and quantified library concentrations using Qubit. Libraries were sequenced with Illumina HiSeq-3000 instrument to an estimated whole-genome coverage of 10-40X (*Supplementary file 5*).

## DNA-sequencing read processing, alignment and SNP discovery

We verified DNA-sequencing read quality using FastQC and trimmed adapter sequences using Trim-Galore. We aligned DNA-seq reads to the stickleback reference genome sequence ('BROAD S1' *Jones et al., 2012*) using *BWA mem* (*Li, 2013*).

We used the Broad Institute Genome Analysis Tool Kit (*GATK*) (*McKenna et al., 2010*) to call Single Nucleotide Polymorphisms (SNPs) in genomic resequencing data, following the DNA-seq best practices (as in June 2016). We ran *GATK HaplotypeCaller* individually for each sample and defining parameters *-stand_call_conf* 30 *-stand_emit_conf* 10 *–emitRefConfidence* GVCF *-variant_index_type* LINEAR *-variant_index_parameter* 128000. The step was followed by joint genotyping using *GenotypeGVCFs* after which we excluded indels from the analysis. The final step was *VariantQualityScoreRecalibration* (*VQSR*). Samples from each ecotype-pair as well as each controlled cross were analyzed together (but separately from other ecotype-pairs or controlled crosses) from *GenotypeGVCFs* -step onwards. Because stickleback lacks a set of known variant sites, we opted for using a hard-filtered set of SNPs as 'true' set of SNPs (with *prior* = 10). We used the *GATK SelectVariants* tool to extract a training set that fulfilled the following thresholds: *QD* >30, *FS* <60, *MQ* >40, *MQRankSum* >−12.5 and *ReadPosRankSum*>-8. After inspection of *VQSR* tranche plots, we selected the 99.9% quality tranche for downstream analysis, which captured 1.66M and 3.52M SNPs with transition/transversion ratios of 1.14 and 1.18 for Tyne and Little Campbell population genomic analyses respectively, and 2.27M, 1.62M, 2.75M, 1.54M SNPs with transitition/transversion ratios ranging from 1.16 to 1.17 for parents of the four crosses (strains from Forss, Tyne, Little Campbell, Shiel) used in allele-specific expression analysis.

## Population genetic analyses

Population genetic analyses were based on a set of six freshwater and six marine fish from both Tyne and Little Campbell River. From the GATK variant calling analysis (described above) we identified over 3.5 million SNPs in the Little Campbell populations and over 1.6 million SNPs in Tyne. We calculated per-site statistics for Weir and Cockerham's $F_{ST}$ (*Weir and Cockerham, 1984*) and average pairwise-nucleotide diversity (Pi) genome-wide, and for 400 kilobase (kb) regions centered on transcription start sites (TSSs) using *VCFtools* (version 0.1.14) (*Danecek et al., 2011*) allowing for a maximum of 4 missing genotypes per SNP for calculation of $F_{ST}$ and a maximum of 2 missing genotypes per SNP for calculation of Pi, corresponding to a maximum of 20% missing genotypes in each case. Negative $F_{ST}$ values were rounded to zero. CSS score was calculated based on Pi and following (*Jones et al., 2012*) in 10 kb non-overlapping windows across the genome. We used the 10 kb windows to assign genome regions as having strongest level of parallel genetic divergence between

Tyne and Little Campbell freshwater and marine ecotypes, keeping the top 1% windows with the highest CSS score.

We used custom R scripts to calculate $F_{ST}$ and Pi in 1 kb windows centered on transcription start sites (TSSs, as reported by *CuffLinks*). We used custom R scripts and the R package *GenomicRanges* to compare the genomic cordinates of transcripts showing parallel expression divergence to the cordinates of the genomic windows showing high CSS values. We calculated the average distance between CSS outlier windows and transcripts in increments of 10 kb and compared the average distance to distances calculated based on 1000 randomized sets of transcripts.

## Allele-specific expression analysis

We defined a set of high-confidence SNPs for allele-specific expression (ASE) analysis, based on genomic resequencing of parent fish used in controlled crosses (above). We then used *GATK FastaAlternateReferenceMaker* to mask the stickleback reference genome in the corresponding position with Ns in order to avoid preferential mapping of reference SNP alleles. We aligned RNA-seq reads from each F1 and the parents onto the N-masked reference genomes using *STAR* and parameters as described above, with the exception that we allowed for only uniquely mapping reads (*–outFilterMultimapNmax* 1). No significant preferential mapping of reference SNPs was observed after these steps.

We selected SNPs where the genotypes of the parents were covered by at least 10 DNA sequencing reads in each parent and where the genotypes of the parents were homozygous for different alleles. These SNP positions were assigned as informative for allele-specific analysis in each cross. Expression levels for allele-specific analyses were represented as read counts overlapping informative SNP positions. We generated allele-specific read counts for F1's and parents with the *GATK ASEReadCounter* tool and enabling default filters (*Castel et al., 2015*). We verified that parents had more than 99% counts assigned to right genotypes and excluded the few SNPs where the RNA-seq reads indicated that both alleles were expressed in a parent that should be homozygous. We combined all individuals in each cross (parents and F1's) in one data frame and normalized read counts between individuals to the total library size using *DESeq2* function *estimateSizeFactors* in order to have equal power across F1s. We then filtered for SNP positions covered by more than 10 reads in at least one F1 in order to avoid underpowered tests of allele-specific expression at loci showing no or very low expression. Finally, we intersected the SNP-based results with transcript models from our reference-guided assembly using the R package *GenomicRanges*, assigning each informative SNP position to an expressed locus and exon.

We tested for ASE in each informative SNP position using a binomial exact test in *R* and an FDR level of 10% (*Storey, 2002*). Normally the null hypothesis of binomial test is 0.5, in our case meaning that 50% of the RNA-seq reads represented either marine or freshwater parent alleles. Our approach takes into account a possible residual effect of preferential mapping of reference allele reads by calculating the null-hypothesis for the binomial test based on the ratio of all reference reads over all alternative reads per each F1 following (*Buil et al., 2015*). The null hypothesis calculated this way was between 0.5 and 0.52, indicating that the residual effect of preferential mapping of reference alleles, if detected, was small. Finally, the results for the ASE test were converted from reference allele versus alternative allele format into marine parent versus freshwater parent by comparing to the genotypes of the parents.

Following ASE testing, we tested for analogous expression difference between parents in the corresponding SNP positions, again using binomial exact test and FDR of 10%. We tested for difference in allele-expression ratio versus parental expression ratio with Fisher's exact test. We then compared ASE significance, ASE sign and ASE magnitude to parental expression difference in order to dissect parental expression differences into divergence classes following (*Landry et al., 2005*), outlined in *Supplementary file 4*. We verified that our results were robust toward sequencing depth by downsampling to 30 million RNA-seq reads per sample (*Figure 3—figure supplement 4*).

Given that the median number of assembled transcript isoforms per gene (locus) is 3, and the mean number of ASE informative SNPs tagging a given gene (locus) range from 2.0 to 6.2, we concluded that the level of evolutionary divergence between marine and freshwater stickleback strains used in our study was insufficient to dissect the *cis-* vs *trans-* genetic architecture underlying expression divergence at the transcript (isoform) level (see Supplemental Note for more information). We

instead analyzed ASE at the gene (locus) level, classifying cis-/trans- architecture of each gene into one divergence class based on the SNP that showed the highest statistical support (see above).

We performed concordance analysis for validating the reproducibility of allele expression levels and divergence classes on SNPs assigned to the same exon and the same transcript (Supplemental Note). For the final classification of transcripts to divergence types, we ordered informative SNPs per transcribed locus per F1 by the product of the p-values of the three tests (above) and selected the SNP that had the lowest product of p-value as a representative SNP for that transcript, per F1. We selected this approach among alternatives after taking into account the high concordance of allele expression levels, relatively complex isoform expression patterns, analysis sensitivity, statistically balanced approach, and parsimony in biological explanation of expression divergence (Appendix 1).

A small number of SNPs showed monoallelic expression in F1's where RNA-seq reads overlapping one of the parent alleles were not observed (e.g. 1129 SNPs in Little Campbell River cross). Analysis of RNA-seq read coverage indicated that these SNP positions had lower coverage specifically of the alternative allele, and this effect was not observed when evaluated based on all SNPs or subsets of SNPs e.g. assigned as cis-diverged. This indicated that the cases of monoallelic expression are likely caused by mappability issues, and that the issue was specific to SNPs showing monoallelic expression. Although including these SNPs did not impact the results in a significant way we decided to exclude SNPs showing monoallelic expression in F1s or parents from the analysis.

We summarized the frequencies of transcripts assigned to divergence classes for each F1 using custom R scripts, excluding transcripts that showed differential expression between sexes at FDR 10%. We then assigned transcripts to two classes according to whether or not they showed parallel expression divergence in PCA and differential expression analysis. We tested for over-representation in divergence classes using a randomization test. For the purposes of calculating the background frequencies of genetic architectures, we defined a set of differentially expressed genes between each ecotype-pair based on PC one loadings (10% outliers, median FDR per outlier 13.5–14.5%, see Supplementary analysis 1).

## Salinity response

A clutch of Tyne marine x freshwater F1 siblings were raised in standard laboratory 3.5 ppt salinity until 3 months old, separated into three groups and acclimated to over 4 months. At reproductive maturity, we analyzed the gill transcriptomes of four F1s from each salinity using RNA-seq.

For testing the effect of salinity acclimation on gene expression in F1 gills, we estimated transcript-level expression using *CuffQuant* and normalized counts for each sample to total library sizes using *CuffNorm*. We then imported the gene count tables into *R* and tested for expression differences between salinity treatments using contrasts and an FDR level of 1%, as implemented in *DESeq2*.

For analysis of expression profiles, we imported FPKM values from *CuffLinks* into R. The FPKM values were highly correlated with normalized expression values from *DESeq2 VarianceStabilizingTransformation*, and allow for a more intuitive interpretation. We sub-set the data to only include the transcripts showing differential expression between at least one contrast and parallel expression divergence (N = 97). We then log-transformed and normalized the expression of each transcript to the average expression level across all samples to produce an expression profile that represent expression in a given sample relative to others (for that transcript). We then compared the profiles of samples with Spearman correlation. A correlation approaching one indicates that expression profiles tended to be similar relative to other samples. In contrast, a correlation approaching −1 indicated that sample profiles were mirror-images of one another. Finally, we clustered the samples based on euclidean distance and visualized the sample similarity profiles using the *pheatmap R* function.

For analysis of allele-specific expression in salinity treatments, we imported allele-specific counts over informative SNPs (as defined above) into R and transformed the counts into log-fold change of marine over freshwater allele. We identified one of the samples acclimated to 35 ppt as having outlier allelic expression levels very different from all other samples and excluded the sample from further analysis. We intersected the SNPs with transcripts that exhibited salinity response and parallel expression divergence (N = 11 transcripts). We measured the similarity of fold-change expression differences between alleles across samples with Spearman correlation, analogous to FPKM counts.

## Additional information

### Funding

| Funder | Grant reference number | Author |
|---|---|---|
| FP7 Ideas: European Research Council | ERC-2013-CoG 617279 | Felicity C Jones |

The funders had no role in study design, data collection and interpretation, or the decision to submit the work for publication.

### Author contributions

Jukka-Pekka Verta, Conceptualization, Formal analysis, Investigation, Visualization, Methodology, Writing—original draft, Project administration, Writing—review and editing, Designed the study, Conducted the experiments, Analyzed the data; Felicity C Jones, Conceptualization, Formal analysis, Supervision, Funding acquisition, Investigation, Visualization, Methodology, Project administration, Writing—draft, review and editing, Designed the study, Contributed to data analysis

### Author ORCIDs

Jukka-Pekka Verta (iD) https://orcid.org/0000-0003-1701-6124
Felicity C Jones (iD) https://orcid.org/0000-0002-5027-1031

### Ethics

Animal experimentation: All of the animals were housed at an approved animal facility and handled according to Baden-Württemberg State approved protocols at the Max Planck Institute for Developmental Biology, Tübingen, Germany (license numbers 35/9185.82-5 and 35/9185.40).

### Decision letter and Author response

Decision letter https://doi.org/10.7554/eLife.43785.037
Author response https://doi.org/10.7554/eLife.43785.038

## Additional files

### Supplementary files

• Supplementary file 1. Sampling locations.
DOI: https://doi.org/10.7554/eLife.43785.024

• Supplementary file 2. RNA-seq library sequencing and yield.
DOI: https://doi.org/10.7554/eLife.43785.025

• Supplementary file 3. Gene Ontology term enrichment. Available as a supplementary xls file. Two different Gene Ontology analyses were performed using parallel divergent gene expression outliers defined through parametric analysis (CuffDiff) and composite PC analysis (compPC). Gene ontologies with significance lower than $1 \times 10^{-5}$ shown.
DOI: https://doi.org/10.7554/eLife.43785.026

• Supplementary file 4. Criteria for defining divergence classes following (*Landry et al., 2005*).
DOI: https://doi.org/10.7554/eLife.43785.027

• Supplementary file 5. DNA sequencing yield.
DOI: https://doi.org/10.7554/eLife.43785.028

• Transparent reporting form
DOI: https://doi.org/10.7554/eLife.43785.029

### Data availability

Data has been deposited to the Sequence Read Archive under the BioProject accession PRJNA530695. All scripts used in data analysis are available at https://github.com/jpverta/verta_

jones_elife_2019.git (copy archived at https://github.com/elifesciences-publications/verta_jones_elife_2019).

The following dataset was generated:

| Author(s) | Year | Dataset title | Dataset URL | Database and Identifier |
|---|---|---|---|---|
| Verta JP, Jones FC | 2019 | Predominance of cis-regulatory changes in parallel expression divergence of sticklebacks | https://www.ncbi.nlm.nih.gov/bioproject/PRJNA530695 | NCBI Sequence Read Archive, PRJNA530695 |

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

## Appendix 1

DOI: https://doi.org/10.7554/eLife.43785.030

### Hierarchical clustering of expression

The gill transcriptome can be characterized by 5 major groups of loci (*Figure 1 – figure supplement 2*) comprising 766 highly expressed genes (2.6%) showing strong enrichment for biological processes including mitochondrial respiration, ATP synthesis coupled proton transport and cytoplasmic translation, 4337 moderately expressed genes (14.8%) enriched for metabolic processes such as mRNA processing and RNA splicing, and functions such as cadherin mediated cell adhesion, 6950 lowly expressed genes (23.7%) enriched with genes involved in protein modification and chromatin modification processes, 5565 low or partially expressed genes (19.0%) enriched in genes involved in developmental processes and regulation of developmental processes, and 11677 predominantly unexpressed genes (39.9%).

### Defining parallel expression divergence using a parametric test

For parametric testing of parallel expression differences between ecotypes, we combined samples from Little Campbell and Tyne and tested for differential expression using *CuffDiff*, specifying the parameter *-dispersion-method* per-condition. We imported the results tables into *R* and selected transcripts that were tested and where the q-value was less than 0.2. We additionally required that the mean expression difference between freshwater and marine ecotypes was of the same sign in Tyne and Little Campbell.

With a false discovery rate of 20%, we identified 120 loci that showed both significant differences in mean and consistent direction (sign) of divergence between ecotype strains from different river systems. The differentially expressed genes tended to have large composite PC loadings (*Figure 1—figure supplement 6*). Top ranking genes include Na-Cl cotransporter (slc12a10), Basolateral Na-K-Cl Symporter (slc12a2), cation proton antiporter 3 (slc9a3.2), Potassium Inwardly-Rectifying Voltage-Gated Channel (kcnj1a.3), potassium voltage-gated channel (KCNA2), Epithelial Calcium Channel 2 (trpv6), Sodium/Potassium-Transporting ATPase (atp1a1.4), aquaporin 3a (aqp3a) — genes known to play a role in osmoregulation in fish and other organisms. This set of differentially expressed loci also include a microRNA (mir-182), 31 loci that are annotated in previous Ensembl gene builds but have unknown function inferred from protein homology to other organisms, and 30 entirely novel loci that have no overlap with gene annotations from Ensembl gene build 90.

We opted to concentrate on the PCA-based results in the main text because we believe the analysis captures better the essence of parallel expression divergence; many small expression differences. We believe the PCA approach is more sensitive in characterizing parallel expression differences mainly because small expression differences would require a much larger sample size to be detected using a parametric linear model.

We used *CuffDiff* also for testing of ecotype-specific expression divergence specifying the same parameters as above. For this analysis, freshwater-marine expression differences were tested separately for Tyne and Little Campbell and the results were compared using *R*. Transcripts that were differentially expressed in both ecotype-pairs with FDR 20% and where the ecotype difference was of the same sign were assigned as 'parallel', whereas if the signs were opposite the transcripts were assigned as 'anti-parallel'.

Overall, 719 differentially expressed transcripts (FDR 20%) were identified using a parametric analysis, the majority of which (N = 515) had marine-freshwater differential expression unique to Little Campbell compared to N = 157 uniquely differential in Tyne strains. Consistent with largely river system-specific ecotype expression divergence only four percent of loci (N = 29) show parallel expression divergence in both rivers (significant differential expression and identical sign of expression difference in both Tyne and Little Campbell) while 2.5% (N = 18) show anti-parallel expression divergence (significant differential expression and opposite sign of expression difference).

## CSS based on Tyne and Little Campbell population data

We calculated a Cluster Separation Score (CSS) in 10 kb non-overlapping windows across the genome following (*Jones et al., 2012*). The CSS score reflects parallel genetic divergence between freshwater and marine fish irrespective of their geographic origin. We assigned the genomic windows with the extreme 0.5% CSS values as regions showing the strongest signal of parallel genetic divergence (449 windows). More than 26 percent of the transcripts evolving in parallel between Tyne and Little Campbell were situated within 10 kb of regions of parallel genetic divergence (randomization test, p<<0.025, *Figure 2—figure supplement 1*), which is two times more than what was observed for the global set of regions showing parallel genetic divergence (main text).

## SNP concordance analysis

We performed SNP concordance analysis to validate the reproducibility of allele expression levels between SNPs assigned to same exons and to different exons of the same transcript. Our assumption for this analysis was that individual SNPs assigned to the same exons should show correlated levels of ASE as well as concordant class of genetic divergence (*cis, trans* etc.) when compared within the same F1 individual. It is worth to note that the divergence class also depend on expression levels assigned to the SNP position in parents, which we ignored for simplicity in this analysis.

SNPs assigned to same exons and showing identical type of genetic divergence (concordant SNPs) tended to have strongly correlated ASE levels (*Appendix 1—figure 1a*). SNPs assigned to same exons but showing different classes of genetic divergence (discordant SNPs) were almost 50% rarer compared to concordant SNPs and, as expected, showed lower level of correlation in ASE and overall smaller allelic differences (*Appendix 1—figure 1b*). Concordant and discordant SNPs within exons showed overall similar distribution among divergence classes, indicating that discordant calls influenced all classes equally (not shown).

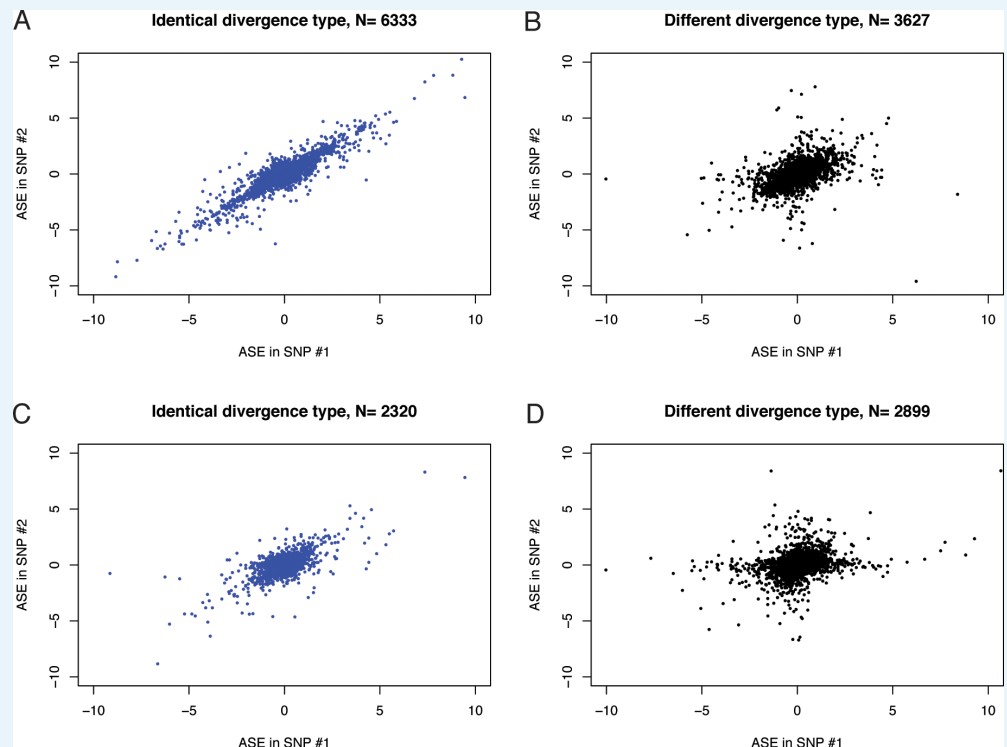

**Appendix 1—figure 1.** Correlation of ASE (log-fold change of RNA-seq counts of marine over freshwater allele) in different pairs of SNPs was used to perform concordance analysis. (**a**) SNPs

assigned to same exons and showing identical type of genetic divergence (concordant SNPs). (**b**) SNPs assigned to same exons but showing different classes of genetic divergence (discordant SNPs). (**c**) SNPs assigned to different exons of the same transcript and showing identical type of genetic divergence. (**d**) SNPs assigned to different exons of the same transcript and showing different type of genetic divergence.

DOI: https://doi.org/10.7554/eLife.43785.031

SNPs assigned to the same transcript but different exons had a correlated level of ASE in cases where the divergence class assigned to both exons were the same (*Appendix 1—figure 1c*). Cases where the exons showed different divergence class showed a lower correlation in ASE (*Appendix 1—figure 1d*). These results are consistent with previous studies demonstrating variable levels of ASE along genes and between exons (*Skelly et al., 2011*). Different exons of the same transcript that show different levels of ASE suggest that ASE effects are specific to single isoforms rather than all isoforms assigned to the same transcript. Through our RNA-seq analysis we identified over 162000 known and new isoforms distributed to 29296 transcribed loci (on average over five isoforms per transcribed locus).

In our final analysis, we opted to classify each transcript into one divergence class, based on the SNP that showed the highest statistical support (see Materials and methods). The justification for this choice was based on the following considerations:

1. Individual SNPs assigned to the same exon tended to show similar ASE levels and divergence types, indicating that dissection of divergence architecture was generally robust to different SNPs within exons. Discordance in divergence types for SNPs assigned to the same exon influenced all divergence types equally and therefore is not expected to bias the results.
2. SNPs assigned to the same transcript but different exons showed different divergence types in roughly half of the cases, and the levels of ASE on the SNP loci were less correlated. This suggests that different exons may experience varying levels of ASE, likely because of alternative isoform expression, as has been demonstrated before (*Skelly et al., 2011*). Visual inspection of expression tracks in candidate genes for variable ASE identified multiple instances of putative alternative isoform expression (example in *Appendix 1—figure 2*). Any procedure that would not distinguish different exons, for example averaging expression levels across SNPs, would therefore suffer from low sensitivity as loci not showing ASE would cancel the signal from loci showing ASE.
3. We discarded the option of averaging ASE levels for SNPs assigned to the same exon because different transcripts and ecotype-pairs showed markedly different densities of SNPs. Averaging would therefore influence transcripts and ecotype-pairs unequally.

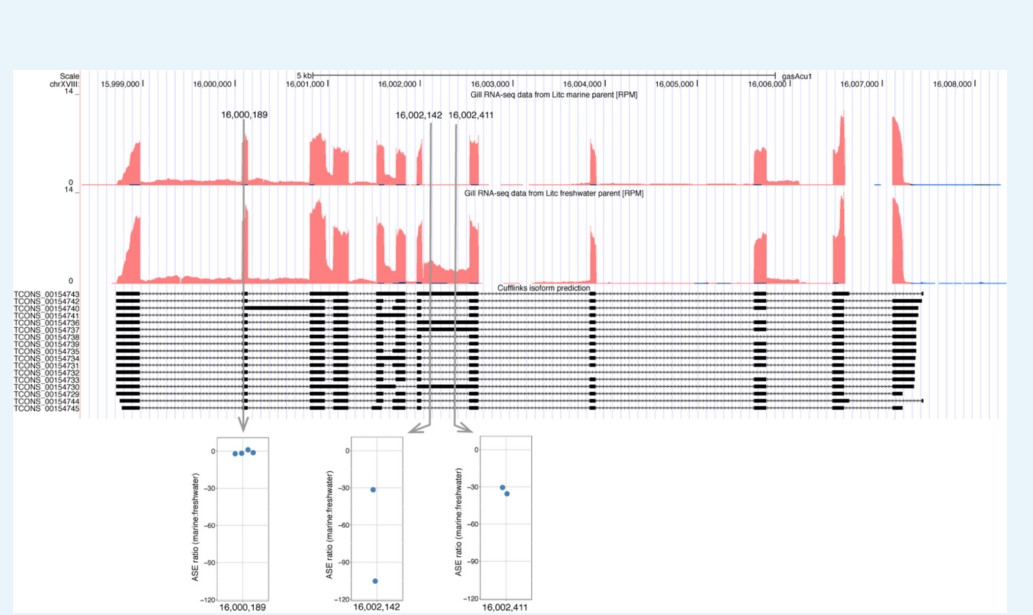

**Appendix 1—figure 2.** Candidate transcript for alternative isoform expression. This example illustrates that SNPs in different regions of the same transcript can show varying levels of ASE associated with alternative splice forms. Expression level is represented by RPM (Reads [mapping to position] Per Million [reads mapped overall]) and corresponds to red track in Little Campbell marine (upper) and freshwater (lower) parents. CuffLinks isoform predictions are represented below expression tracks. The marine and freshwater alleles are distinguished by three SNPs mapping to two exons. SNP at position chrXVIII:16000189 shows similar expression level of both alleles and hence no ASE. SNPs at positions 16002142 and 16002411 map to an alternatively spliced region of the transcript where expression is observed in freshwater allele but not in marine allele, and hence the SNPs show ASE towards freshwater.
DOI: https://doi.org/10.7554/eLife.43785.032

## Analysis of power for *cis/trans* test

In the approach we employed for testing the genetic architecture of expression variation (*Landry et al., 2005*), expression difference is tested between alleles of F1's and between parent fish using a binomial test. Then the expression ratios of alleles (within F1's) are compared to the expression ratios of the alleles between the parents.

- *Cis* regulation necessitates that there is expression difference between the alleles within F1's as well as between parents, and that the ratio of allele 1 versus allele two expression is conserved in F1's and parents.
- *Trans* regulation necessitates that there is expression difference between the parents and that the ratios of allele 1 versus allele two expression is not conserved between the parents and F1's (this is the definition of *trans* - that is expression difference not co-segregating with alleles (*Wittkopp et al., 2004*).

We used simulated data and estimated the power to detect genetic effects under two scenarios:

- all divergence between parents is in *cis*
- all divergence between parents is in *trans*

By simulating count differences between parents based on the distribution of observed data, we defined threshold values for minimal parental difference where *cis* and *trans* tests result in significant outcomes. The difference in power proved out to be small; *cis* test would result in significant outcome with a minimum parental difference of 5 counts, *trans* test with seven counts. Theoretically there is a gap in parental difference between 5–7 counts where a

*cis* test could result in significant outcome but *trans* test would fail to detect differences. In our observed data, 1144 genes fall into this category in Tyne and 1402 in Little Campbell. These genes are assigned mainly into the 'conserved' category of divergence classes. The large majority of genes assigned as *cis* or *trans* have higher parental difference than 5 or 7, indicating that the genes falling into the category of 5–7 parental difference are indeed 'conserved' in regulation and that the difference in power does not influence the overall frequencies of *cis* and *trans* regulated genes. Indeed, we observe from the attached analysis that *trans*-regulated genes tend on average to have larger parental difference compared to *cis*-regulated genes, and that this difference is observed with parental differences where our tests have equal power to detect differences (parental difference >>7).

## Comparison of subtraction-based methods to estimate the magnitude of *cis* and *trans* divergence

We use a variant of the subtraction method to calculate average *cis* and *trans* divergence in the analysis in **Figure 5**. Overall, the categorization method of **Landry et al. (2005)** and the subtraction method result in high concordance; genes categorized as *cis* based on **Landry et al. (2005)** test show high *cis* estimates and low *trans* estimates using the subtraction method, and vice versa for genes assigned as *trans*.

Some problems with subtraction-based approaches has been articulated in a recent publication (**Fraser, 2019**). In summary, when estimating *trans* divergence based on *cis*-acting variation, statistical errors in *cis*-estimation can be automatically negatively correlated with the errors in *trans*-estimation. If this is the case, we expect that the estimates for *cis* and *trans* divergence would be negatively correlated with each other. **Fraser (2019)** suggests a simple solution to this problem: by estimating *cis*-divergence in one F1 and *trans*-divergence in another, the errors are uncorrelated and the method is unbiased.

We compared our subtraction method used in **Figure 5** to the one proposed by Fraser (called 'cross replicate correlation'). For genes assigned as *cis* or *trans*, both methods showed low correlation between *cis* and *trans* estimates for individual genes (which is desired). Estimates of *cis* and *trans* divergence were highly correlated using the two approaches (Supplementary analysis 3).

