## [Decision Letter]

Thank you for submitting your article "Predominance of *cis*-regulatory changes in parallel expression divergence of sticklebacks" for consideration by *eLife*. Your article has been reviewed by three reviewers and the evaluation has been overseen by a Reviewing Editor and Diethard Tautz as the Senior Editor. The reviewers have opted to remain anonymous.

The reviewers have discussed the reviews with one another and the Reviewing Editor has drafted this decision to help you prepare a revised version of your manuscript for resubmission.

Summary:

This manuscript describes an original and informative study characterizing the extent of parallel expression divergence in stickleback gills and its genetic architecture. Using a mixture of parental originating from freshwater and marine environments and some F1 hybrids, the authors describe the *cis-/trans*- acting basis of transcription change, showing that *cis*-acting change overlap with genomic regions carrying population genetics signatures of adaptation. They further investigate patterns of dominance, epistasis and plasticity affecting genes with parallel expression divergence and conclude that *cis*-regulatory variants are best suited to be the preferred basis of adaptive evolution in gene transcription levels. Finally, the authors observe that *cis*-regulation makes a predominant contribution to the genetic assimilation of plastic responses into heritable variation. In this study, genes were also identified that are excellent candidates for evolved adaptive changes in *cis*-regulation that can be the subject of future studies.

The stickleback system is arguably the best platform to investigate the role of gene expression variation in parallel adaptation because of the many examples of repeated adaptation to freshwater as well as the large body of previous work documenting signatures of selection at the genomic level. This novel piece of work informs the community about the pattern and process by which gene expression changes contribute to adaptive phenotypic change. We therefore believe it has the potential to be eventually suitable for publication in *eLife*. However, we have noted three crucial shortcomings that should be addressed in a much revised version as well as two important aspects that are not sufficiently considered in the present manuscript. These can be summarized in 5 major points, three of which warrant essential revisions.

Essential revisions:

Point 1. The analysis of enrichment in *cis*-regulatory variation among genes showing parallel expression divergence was not properly tested. While *cis*-regulatory variation appears to be pervasive, and therefore an important component of the adaptation of gene expression, the question of the universe against which this enrichment is tested is crucial. The authors tested the proportion of *cis*-regulatory changes detected in genes with parallel expression change against the whole set of genes with analyzable ASE (e.g. expressed and with SNPs allowing assignment of parental origin), most of which show no parental difference. Instead, we believe the right test would be to compare parallel-divergence genes to random genes with a significant parental expression difference (regardless of whether it is parallel or not). Since the conclusion that parallel divergent genes are more *cis* than random genes is the titular claim of the paper, this question is crucial and the authors may have to re-formulate their conclusions.

Point 2. The unconventional use of PCA to identify the parallel-diverged genes also raised questions. We wondered why you did not use a linear model of the type: expression ~ river x ecotype or expression ~ ecotype/river. The robustness of your approach against random patterns of gene expression variation is not known. Choosing 1% of the tail of the distribution is a rather blind approach that could be better supported with a permutation approach. We believe you should produce an estimate of the proportion of false positives among the 586 genes you identify as showing a parallel expression change. Here again, depending on the proportion of false positives, your conclusions may have to be revised.

Point 3. An inherent difficulty in *cis-/trans*- studies is that the two sources of variation differ in the respective power with which significant differences can be identified. In F1, alleles are always expressed in the same cell, by contrast with adult parental fishes. *Cis-/trans* frequencies can be computed. A potential solution is to infer the proportion of parental expression variance that is associated with a detectable *cis*-acting change in hybrids.

Point 4. The possibility that epigenetic differences could explain some proportion of the ASE reported here should be discussed, since the parents of the F1 grew up in the wild.

Point 5. Finally, we wondered whether the authors have deliberately chosen to omit references to studies of gene expression completed in plant systems. Back in 2014, a study conducted in maize reported that *cis-*acting variants are enriched in genomic segments carrying signatures of selection (Lemmon et al., 2014), in *Arabidopsis* species specific functions were enriched in derived *cis*-acting changes (He et al., 2016) and in Capsella grandiflora, *cis*-acting changes were associated with relaxed selection (Steige et al., 2017). It seems the authors should refer to the broader body of literature dissecting the genetic basis of expression changes associated with adaptation. Placing their finding into the broader context of the field will give greater resonance to this work, beyond the boundaries of the fish community.

---

## [Author Response]

Essential revisions:Point 1. The analysis of enrichment in cis-regulatory variation among genes showing parallel expression divergence was not properly tested. While cis-regulatory variation appears to be pervasive, and therefore an important component of the adaptation of gene expression, the question of the universe against which this enrichment is tested is crucial. The authors tested the proportion of cis-regulatory changes detected in genes with parallel expression change against the whole set of genes with analyzable ASE (e.g. expressed and with SNPs allowing assignment of parental origin), most of which show no parental difference. Instead, we believe the right test would be to compare parallel-divergence genes to random genes with a significant parental expression difference (regardless of whether it is parallel or not). Since the conclusion that parallel divergent genes are more cis than random genes is the titular claim of the paper, this question is crucial and the authors may have to re-formulate their conclusions.

We begin by summarising the motivation for this test: we are asking what the relative proportion of different regulatory categories underlying parallel gene expression divergence, compared to the overall frequencies of those regulatory categories.

The reviewers recommend the background comprise “random genes with a significant parental expression difference (regardless of whether it is parallel or not)”. We understand the concern raised by the reviewers however, we do not believe that the test proposed in the review the is the best way to proceed. As per our test (please see Supplementary file 4), the proposed procedure leads to an undesirable situation where the background set includes all genes except those assigned to “conserved” or “compensatory” (since by definition these classes show no parental difference). Such a background set would not align with the intended purpose of the analysis, as we are interested in knowing what is the over-representation of regulatory categories in parallel diverged genes above the overall genetic architectures?

To illustrate this point we have performed the suggested analysis using the background set as per comment 1, and arrive at the results shown in Author response image 1.

**Author response image 1. respfig1:** Genes with parallel marine-freshwater divergence in expression are enriched in cis-divergence. Overrepresentation of expression regulation types associated with parallel evolving transcripts, compared to genes with significant parental expression divergence (regardless of whether it is parallel or not). Points represent the observed values in four F1's per cross. Box-plots represent the background expectation.

The first thing to draw your attention to is that the background distributions are no longer meaningful for “compensatory” and “conserved” classes (box plots at zero) because the background set is required to contain genes that show significant expression divergence between the parents, which by definition these categories do not contain. Secondly, in the Tyne, genes with parallel expression divergence are not over-represented in *cis*, but in genes assigned to “conserved”, as well as those assigned as “compensatory” classes. As noted above, the comparison is invalid because by definition the background set cannot contain genes assigned as “conserved” or “compensatory”.

To accommodate the recommendation that the background set of genes to which genes with parallel expression divergences are compared should comprise only genes that are differentially expressed between ecotypes, we re-analyzed the data following the steps below:

i) We performed principal components analyses (PCAs) between freshwater and marine ecotypes separately for Little Campbell and Tyne.

ii) We defined a set of differentially expressed genes between ecotypes based on principal component 1 loadings, which is the major principal component separating the ecotypes (10% outliers, median FDR 13.5-14.5% (Tyne and Little Campbell respectively), see response 2 for calculation).

iii) We used the principal component 1 outliers, which are differentially expressed between ecotypes (and therefore fulfil the requirement in the reviewer’s comment), as the background set of genes to which genes with parallel expression divergence were compared.

Using set of differentially expressed genes between ecotypes as the background set, we provide the results in Figure 4, which leads to similar conclusion as in our initial analysis: genes with parallel expression divergence are enriched with cis-regulation compared to the background set of diverged genes.

The above analysis has now been included in the main text (replacing the original analysis). We have modified the text and Figure 4 to accommodate these changes.

Point 2. The unconventional use of PCA to identify the parallel-diverged genes also raised questions. We wondered why you did not use a linear model of the type: expression ~ river x ecotype or expression ~ ecotype/river. The robustness of your approach against random patterns of gene expression variation is not known. Choosing 1% of the tail of the distribution is a rather blind approach that could be better supported with a permutation approach. We believe you should produce an estimate of the proportion of false positives among the 586 genes you identify as showing a parallel expression change. Here again, depending on the proportion of false positives, your conclusions may have to be revised.

It is a misunderstanding that we did not use a linear model to identify genes with parallel expression divergence. We have indeed performed such analysis and refer to it in the results (full analysis in the supplementary note under the section: Defining parallel expression divergence using a parametric test). The genes identified using the linear model largely overlaps the set of genes identified using PCA (Figure 1—figure supplement 6).

We opted to concentrate on the PCA results in the main text for the following reasons:

i) We believe the PCA results are more sensitive than linear modelling.

ii) We believe PCA captures better the essence of parallel expression divergence (many small expression differences).

Our understanding is that PCA is more sensitive in characterizing parallel expression differences mainly because small expression differences would require a much larger sample size to be detected using a linear model. We corrected the text emphasizing that the fact that the two approaches give concordant results.

Following the reviewers' suggestion, we calculated an FDR for the parallel expression divergence we observed (result: median FDR = 2.6% ). Since we are unaware of any standard way to test the significance of single PCA loadings in the context of the overall PCA, we used an approach inspired by: Linting, van Os and Meulman, (2011).

The procedure has the following steps:

i) For each outlier gene, shuffle the sample labels while keeping all other data as original.

ii) Re-perform PCA and calculate the composite PC axes describing significant ecotype divergence.

iii) Store the composite PC loadings from the permutation.

iv) Perform steps *i-iii* for all possible combinations of dividing the 16 samples to two groups of 8 samples (6435 unique combinations) for each composite PC outlier to produce a null distribution of composite PC loadings.

v) Calculate Z-score and P-value for observing a loading higher than the permuted null.

vi) Correct for multiple testing (FDR).

The procedure tests the null hypothesis that the composite PC loading of a given gene is not significantly higher compared to composite PC loadings calculated from non-structured data (i.e. where the sample identities have been shuffled). This approach produces an FDR for each composite PC outlier. In the main text, we report the median FDR over all composite PC outliers (2.6% ).

Figure 1—figure supplement 5 has been added to the revised manuscript. Panel a represents sorted composite PC loading for all genes and illustrate 1% cutoffs for extreme values with a dotted line. Panel b represents the FDR values for the 586 transcripts designated as composite PC outliers.

This information was added to the manuscript.

Point 3. An inherent difficulty in cis-/trans- studies is that the two sources of variation differ in the respective power with which significant differences can be identified. In F1, alleles are always expressed in the same cell, by contrast with adult parental fishes. Cis-/trans frequencies can be computed. A potential solution is to infer the proportion of parental expression variance that is associated with a detectable cis-acting change in hybrids.

The reviewers point out a difference in power that exists when *cis* regulation is calculated from allelic ratios within offspring and *trans* regulation from parental ratios minus *cis*. This can make comparisons between *cis* and *trans* (e.g. the relative proportion of *cis*- and *trans*-) problematic. The reviews suggest using a model to estimate the proportion of variance in parental expression difference that can be explained by *cis*-acting differences in F1s.

With the exception of the analysis in Figure 5, we estimate *cis* and *trans* divergence based on an alternative approach that has been widely used, and where one does not use subtraction to estimate *trans* divergence (Hartl et al., (2005). Also note that based on a power analysis of simulated data, we observe very little difference in statistical power between (this alternative) *cis* and *trans* tests.

We use a variant of the subtraction method to calculate average *cis* and *trans* divergence in the analysis in Figure 5. In this analysis we do not make comparisons between *cis*- and *trans*- categories but rather within each category study the parallelism (correlation) in magnitude of their effects among different populations. For this reason, we believe the analysis and conclusions presented in the analyses in Figure 5 are unaffected by potential biases in the differential power to detect *cis* and *trans* regulation.

Overall, the categorization method of Landry et al., and the subtraction method result in high concordance; genes categorized as *cis* based on Landry et al. test show high *cis* estimates and low *trans* estimates using the subtraction method, and vice versa for genes assigned as *trans*.

Some problems with subtraction-based approaches has been articulated in a recent publication (Fraser, (2019). In summary, when estimating *trans* divergence based on *cis*-acting variation, statistical errors in *cis*-estimation can be automatically negatively correlated with the errors in *trans*-estimation. If this is the case, we expect that the estimates for *cis* and *trans* divergence would be negatively correlated with each other. Fraser, (2019) suggests a simple solution to this problem: by estimating *cis*-divergence in one F1 and *trans*-divergence in another, the errors are uncorrelated and the method is unbiased.

We compared our subtraction method used in Figure 5 to the one proposed by Fraser (called “cross replicate correlation”). For genes assigned as *cis* or *trans,* both methods showed low correlation between *cis* and *trans* estimates for individual genes (which is desired). Estimates of *cis* and *trans* divergence were highly correlated between the two approaches.

We have added this information to Appendix 1.

Point 4. The possibility that epigenetic differences could explain some proportion of the ASE reported here should be discussed, since the parents of the F1 grew up in the wild.

This is a misunderstanding. The parents were indeed raised in the lab in identical environment to the F1’s, as explained in the Materials and methods section. We revised the Materials and methods section for clarity.

Point 5. Finally, we wondered whether the authors have deliberately chosen to omit references to studies of gene expression completed in plant systems. Back in 2014, a study conducted in maize reported that cis-acting variants are enriched in genomic segments carrying signatures of selection (Lemmon et al., 2014), in Arabidopsis species specific functions were enriched in derived cis-acting changes (He et al., 2016) and in Capsella grandiflora, cis-acting changes were associated with relaxed selection (Steige et al., 2017). It seems the authors should refer to the broader body of literature dissecting the genetic basis of expression changes associated with adaptation. Placing their finding into the broader context of the field will give greater resonance to this work, beyond the boundaries of the fish community.

We regret that we had missed references to this valuable literature and have added discussion of these studies to the manuscript.